# Creatinine versus cystatin C for renal function-based mortality prediction in an elderly cohort: The Northern Manhattan Study

**Joshua Z. Willey**[1]*, **Yeseon Park Moon**[1], **S. Ali Husain**[2], **Mitchell S. V. Elkind**[1,3], **Ralph L. Sacco**[4], **Myles Wolf**[5], **Ken Cheung**[6], **Clinton B. Wright**[4], **Sumit Mohan**[2,3]

**1** Division of Nephrology, Vagelos College of Physicians and Surgeons, Columbia University Medical Center, New York, NY, United States of America, **2** Department of Neurology, Vagelos College of Physicians and Surgeons, Columbia University, New York, NY, United States of America, **3** Department of Epidemiology, Mailman School of Public Health, Columbia University, New York, NY, United States of America, **4** Departments of Neurology and Public Health Sciences, Leonard M. Miller School of Medicine, the McKnight Brain Institute and the Neuroscience Program, University of Miami, Miami, FL, United States of America, **5** Division of Nephrology, Department of Medicine, Duke University School of Medicine, Durham, NC, United States of America, **6** Department of Biostatistics, Mailman School of Public Health, Columbia University, New York, NY, United States of America

* jzw2@cumc.columbia.edu

**Data Availability Statement:** The full dataset cannot be shared publicly because of protected health information (PHI). A dataset with all the variables used in the analysis, except for PHI, is

## Abstract

### Background

Estimated glomerular filtration rate (eGFR) is routinely utilized as a measure of renal function. While creatinine-based eGFR (eGFRcr) is widely used in clinical practice, the use of cystatin-C to estimate GFR (eGFRcys) has demonstrated superior risk prediction in various populations. Prior studies that derived eGFR formulas have infrequently included high proportions of elderly, African-Americans, and Hispanics.

### Objective

Our objective as to compare mortality risk prediction using eGFRcr and eGFRcys in an elderly, race/ethnically diverse population.

### Design

The Northern Manhattan Study (NOMAS) is a multiethnic prospective cohort of elderly stroke-free individuals consisting of a total of 3,298 participants recruited between 1993 and 2001, with a median follow-up of 18 years.

### Participants

We included all Northern Manhattan Study (NOMAS) participants with concurrent measured creatinine and cystatin-C.

available from the Columbia University Academia commons site (https://academiccommons. columbia.edu) where any researcher can obtain it.

**Funding:** This work was supported by: MSVE, RLS: NINDS R01 NS029993. The funders had no role in study design, data collection and analysis, decision to publish, or preparation of the manuscript.

**Competing interests:** The authors have declared that no competing interests exist.

## Main measures

The eGFRcr was calculated using the CKD-EPI 2009 equation. eGFRcys was calculated using the CKD-EPI 2012 equations. The performance of each eGFR formula in predicting mortality risk was tested using receiver-operating characteristics, calibration and reclassification. Net reclassification improvement (NRI) was calculated based on the Reynolds 10 year risk score from adjusted Cox models with mortality as an outcome. The primary hypothesis was that eGFRcys would better predict mortality than eGFRcr.

## Results

Participants (n = 2988) had a mean age of 69±10.2 years and were predominantly Hispanic (53%), overweight (69%), and current or former smokers (53% combined). The mean eGFRcr (74.68±18.8 ml/min/1.73m$^2$) was higher than eGFRcys (51.72±17.2 ml/min/1.73m$^2$). During a mean of 13.0±5.6 years of follow-up, 53% of the cohort had died. The AUC of eGFRcys (0.73) was greater than for eGFRcr (0.67, p for difference<0.0001). The proportions of correct reclassification (NRI) based on 10 year mortality for the model with eGFRcys compared to the model with eGFRcr were 4.2% (p = 0.002).

## Conclusions

In an elderly, race/ethnically diverse cohort low eGFR is associated with risk of all-cause mortality. Estimated GFR based on serum cystatin-C, in comparison to serum creatinine, was a better predictor of all-cause mortality.

## Introduction

The prevalence of chronic kidney disease (CKD) increases dramatically among the elderly [1, 2] and has been identified by several investigators as a risk factor for cardiovascular disease (CVD) related outcomes including mortality [3, 4], heart failure, myocardial infarction [5], stroke [6], and cognition [7–9]; it is furthermore linked to frailty [10, 11]. The impact of CKD on CVD outcomes is independent of their shared risk factors, such as hypertension and diabetes, and in excess of other known risk factors including prevalent CVD [5, 12]. In addition, the increased mortality observed in diabetic patients is predominantly accounted for by the presence of CKD [12]. Furthermore, CKD has a disproportionate burden among those with lower socio-economic status, blacks and Hispanics [13], and may partly explain the increased medication adverse events seen in elderly blacks and Hispanics [14, 15]. Despite these well-documented consequences of CKD, there is a paucity of data in elderly diverse cohorts on the prevalence of CKD as well as the impact of CVD. Furthermore, it is not well known if in elderly diverse populations estimated glomerular filtration (eGFR) calculations using either serum creatinine or cystatin-C can adequately predict CVD and mortality. Previously we and others have shown that eGFR equations using creatinine or cystatin-C can provide significantly divergent estimates of the prevalence of CKD [16]. The goals of this study were to examine 1) the association of CKD using eGFR from creatinine (eGFR$_{cr}$) or eGFR from cystatin-C (eGFR$_{cys}$)with CVD and mortality in an elderly race/ethnically diverse cohort, and 2) performance of eGFR$_{cr}$ and eGFR$_{cys}$ in predicting mortality risk. We hypothesized that a eGFR$_{cys}$ would predict risk of mortality more accurately compared to eGFR$_{cr}$.

## Methods

### Recruitment of the cohort

The recruitment and assessment of the Northern Manhattan Study (NOMAS) cohort has been described in previous publications [17]. Briefly, eligible participants were: 1) stroke free; 2) resident of at least 3 months duration of Northern Manhattan as defined by zip-codes 10031, 10032, 10033, 10034, & 10040; 3) randomly derived from a household with a telephone; 4) age 40 years or older (changed to age 55 or older in 1998) at the time of first in-person assessment. Participants were recruited between 1993–2001and followed longitudinally to present date. All participants gave informed consent to participate in the study. Race-ethnicity was determined by self-identification and standardized questions were used regarding hypertension, diabetes, cigarette smoking, alcohol intake and cardiac comorbidities. Blood pressure was measured twice, before and after each examination, and averaged. Hypertension was defined as a blood pressure ≥140/90 mmHg, the patient's self-report of hypertension, or use of anti-hypertensive

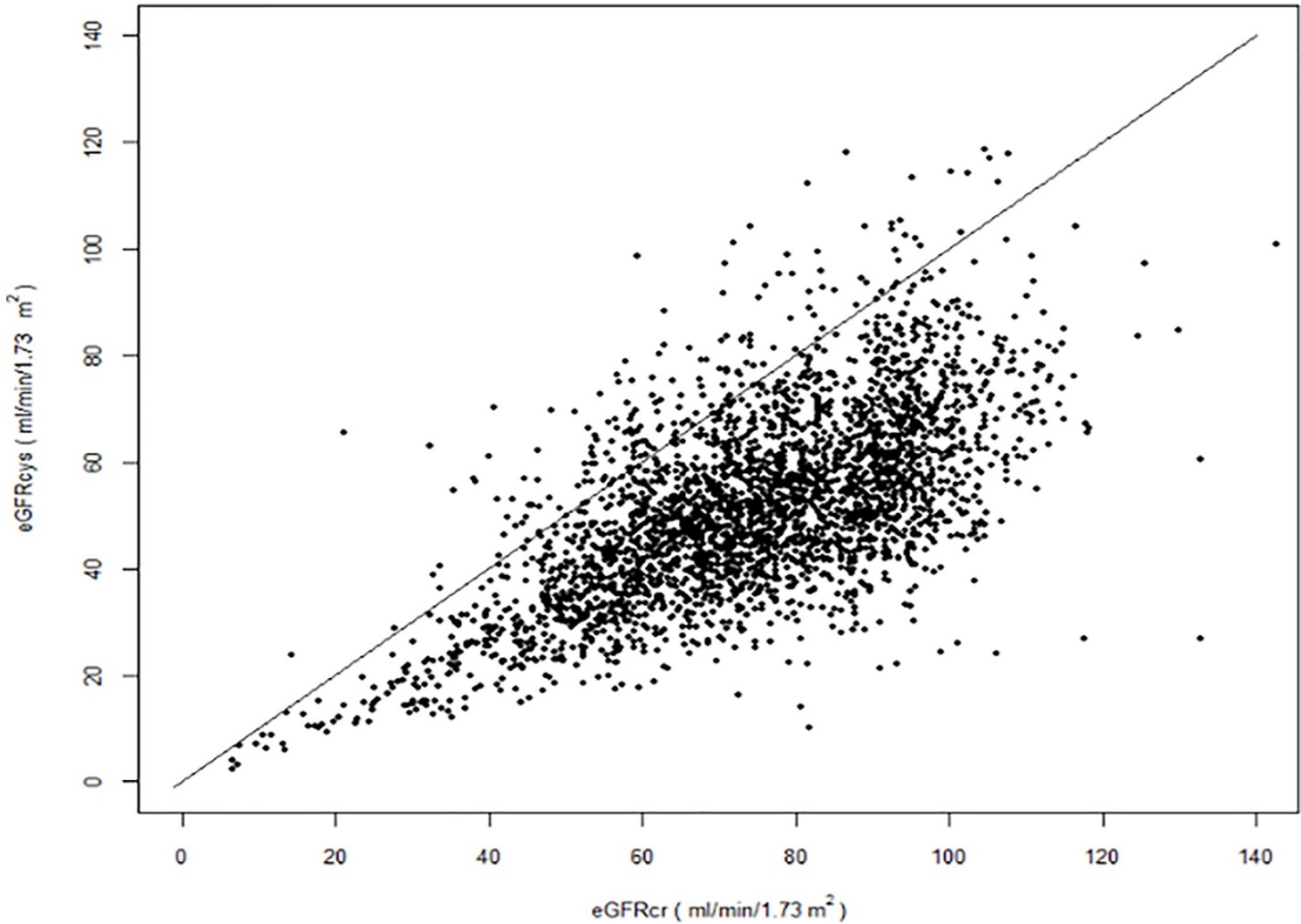

**Fig 1. Dot plot of estimated glomerular filtration rate using serum creatinine and cystatin-C.**

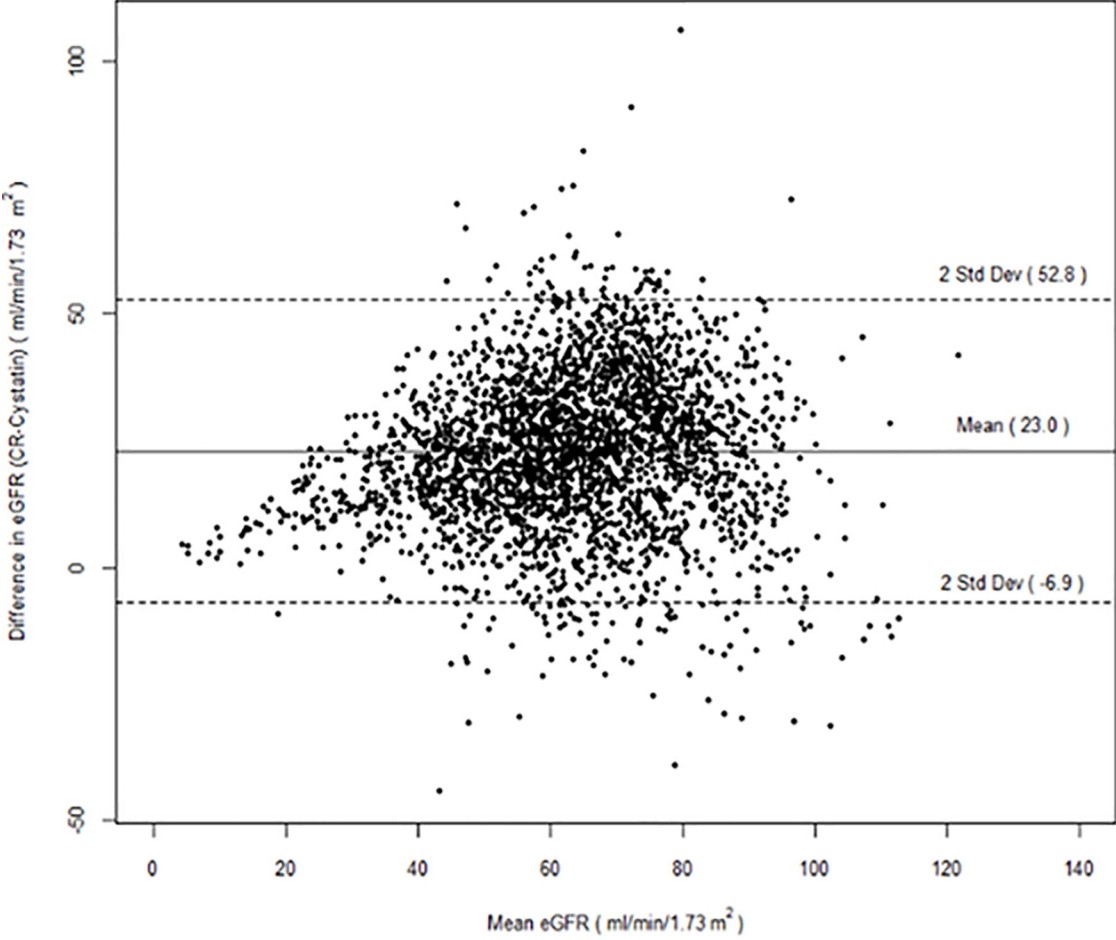

**Fig 2. Bland-Altman plot of estimated glomerular filtration rate using serum creatinine and cystatin-C.**

medications. Diabetes mellitus was defined by the patient's self-report of a history of diabetes, use of insulin or oral anti-diabetic medication, or fasting glucose ≥126 mg/dl. Hypercholesterolemia was defined as having a total cholesterol level of greater than 200 mg/dl, use of cholesterol lowering medications, or self-reported history of hypercholesterolemia. The study was approved by the institutional review boards of Columbia University Irving Medical Center and the University of Miami.

### Measurement of creatinine and cystatin-C

Blood samples were obtained during baseline enrollment from 1993–2001. All laboratory testing was performed at Columbia University Medical Center or at the University of Miami. Serum creatinine (mg/dL) was measured using Olympus instrumentation with a Jaffe-based method. Although the initial creatinine concentrations were measured prior to the isotope dilution mass spectroscopy (IDMS) standardization for estimated GFR, creatinine was re-measured in 100 samples stored at -80˚C using an IDMS-traceable method for creatinine measurement in order to develop a correction factor similar to what had been done successfully by other cohorts [18] [19]. The mean difference between standardized and non-standardized

**Table 1. Baseline demographics of the Northern Manhattan Study.**

|  | Mean (standard deviation) or No. (proportion as %) |
|---|---|
| Age, years, mean (standard deviation) | 69 (10.2) |
| Male | 1101 (37%) |
| Non-Hispanic black | 725 (24%) |
| Non-Hispanic white | 619 (21%) |
| Hispanic | 1577 (53%) |
| Education (completed high school) | 1377(46%) |
| Medicaid/no insurance | 1287 (43%) |
| Diabetes | 634 (21%) |
| Hypertension | 2196 (74%) |
| Body-mass index, mean (std) | 27.8 (5.5) |
| Active tobacco | 498 (17%) |
| Prior tobacco use | 1084 (36%) |
| Hypercholesterolemia | 1893 (63%) |
| Heart disease | 704 (24%) |
| Serum creatinine, mg/dL | 0.96 (0.4) |
| Serum cystatin C, mg/L | 1.4 (0.6) |
| eGFRcr ml/min/1.73m$^2$ | 74.68 (18.8) |
| eGFRcys ml/min/1.73m$^2$ | 51.72 (17.2) |

**Table 2. Associations of estimated glomerular filtration using serum creatinine and cystatin-C with mortality and vascular outcomes in the Northern Manhattan Study.**

|  | eGFRcr < 60 ml/min/1.73m$^2$ (unadjusted hazards ratio, 95% confidence interval) | eGFRcr < 60 ml/min/1.73m$^2$ (adjusted hazards ratio, 95% confidence interval)* | eGFRcys < 60 ml/min/1.73m$^2$ (unadjusted hazards ratio, 95% confidence interval) | eGFRcys < 60 ml/min/1.73m$^2$ (adjusted hazards ratio, 95% confidence interval)* | eGFRcr (per 10ml/min/1.73m$^2$ increase) (unadjusted hazards ratio, 95% confidence interval) | eGFRcys (per 10ml/min/1.73m$^2$ increase) (unadjusted hazards ratio, 95% confidence interval) | eGFRcr (per 10ml/min/1.73m$^2$ increase) * (adjusted hazards ratio, 95% confidence interval) | eGFRcys (per 10ml/min/1.73m$^2$ increase) * (adjusted hazards ratio, 95% confidence interval) |
|---|---|---|---|---|---|---|---|---|
| All-cause mortality | 2.21, 1.99–2.46 | 1.24, 1.11–1.39 | 2.74, 2.41–3.12 | 1.41, 1.22–1.63 | 0.79, 0.76–0.81 | 0.64, 0.62–0.67 | 0.94, 0.91–0.97 | 0.80, 0.77–0.83 |
| Vascular Death | 2.17, 1.78–2.64 | 1.16, 0.94–1.44 | 3.00, 2.38–3.79 | 1.45, 1.11–1.88 | 0.80, 0.76–0.84 | 0.64, 0.60–0.68 | 0.97, 0.92–1.03 | 0.82, 0.76–0.88 |
| Non-vascular death | 2.01, 1.73–2.34 | 1.20, 1.02–1.41 | 2.41, 2.02–2.87 | 1.35, 1.11–1.65 | 0.80, 0.77–0.83 | 0.66, 0.63–0.69 | 0.94, 0.89–0.98 | 0.80, 0.75–0.84 |
| Stroke | 1.76, 1.39–2.22 | 1.17, 0.90–1.51 | 1.51, 1.19–1.91 | 0.98, 0.74–1.28 | 0.85, 0.80–0.90 | 0.83, 0.78–0.89 | 0.95, 0.89–1.02 | 0.99, 0.91–1.07 |
| Myocardial infarction | 2.24, 1.67–3.02 | 1.50, 1.09–2.06 | 2.03, 1.47–2.81 | 1.38, 0.96–2.00 | 0.79, 0.74–0.86 | 0.73, 0.67–0.80 | 0.89, 0.81–0.96 | 0.83, 0.74–0.93 |
| Combined vascular endpoint** | 2.08, 1.77–2.45 | 1.22, 1.02–1.46 | 2.28, 1.92–2.72 | 1.28, 1.05–1.57 | 0.82, 0.79–0.85 | 0.71, 0.67–0.74 | 0.97, 0.92–1.01 | 0.88, 0.83–0.93 |

*Adjusted for age, sex, race-ethnicity, education, Medicaid/no insurance, diabetes, hypertension, body-mass index, tobacco use, hypercholesterolemia, and heart disease
**Stroke, myocardial infarction, vascular death

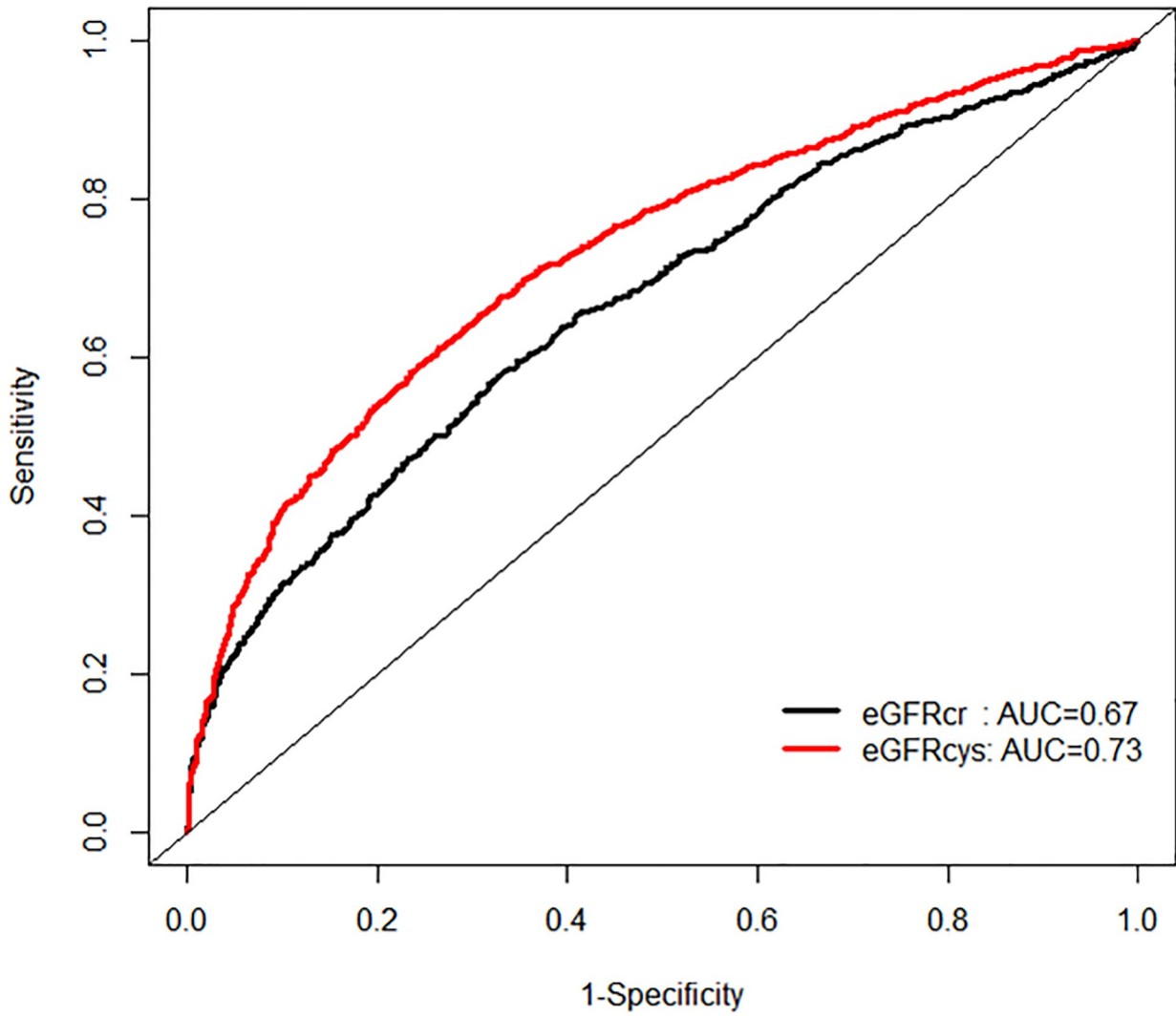

**Fig 3. Area under the curve (AUC) comparisons for model fit for all-cause mortality of glomerular filtration rate estimated by serum creatinine (eGFRcr) and cystatin-C (eGFRcys) at five and ten years of follow up.**

creatinine was -0.056 ± 0.079 mg/dL. In the absence of a meaningful difference, a calibration factor was not applied prior to using the creatinine for GFR estimation using the CKD-EPI 2009 equation [20]. However, a sensitivity analysis was performed by repeating the primary analysis using creatinine values after calibration factor application. Cystatin-C (mg/L) was measured on samples (84% plasma, 14% serum, 2% unspecified) stored at -80˚C using Roche Diagnostics Cystatin Reagents on a Roche analyzer, standardized against ERM-DA471/IFCC reference material (intra-assay coefficient of variation (CV) of 2.8% and an inter-assay CV of 4.1%; reference range 0.5–1.3 mg/L). Cystatin-based GFR estimation used the CKD-EPI 2012 equation [21].

## Statistical analysis

The primary outcome of interest was all-cause mortality, with secondary outcomes of vascular mortality, non-vascular mortality, stroke, MI, and a combined vascular outcome (stroke, MI,

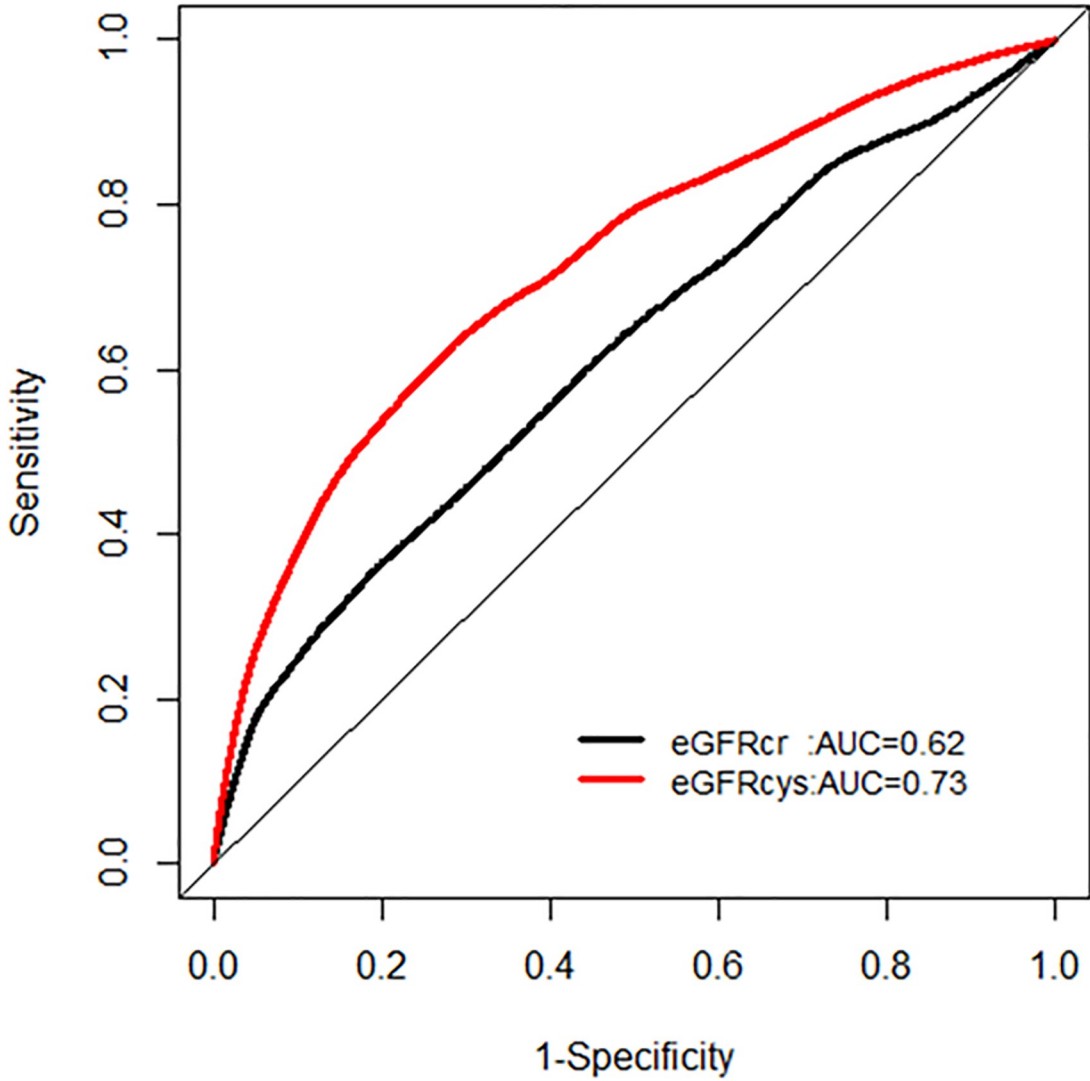

**Fig 4. Comparison of area under the curve for all-cause mortality at 5 years for glomerular filtration rate estimated by serum creatinine (eGFRcr) and cystatin-C (eGFRcys).**

vascular death). The association of eGFR, defined as $< 60$ ml/min/1.73m$^2$ and per 10 ml/min/1.73m$^2$, with the outcomes in this study was examined using Cox proportional hazard models to calculate hazards ratios (HR) and 95% confidence intervals (CI). The models were first calculated unadjusted and then followed by adjusting for cardiovascular disease risk factors (age, sex, race-ethnicity, education, Medicaid/no insurance, diabetes, hypertension, body-mass index, tobacco use, hypercholesterolemia, and heart disease). In order to examine the performance in mortality risk prediction for eGFRcr and eGFRcys, we constructed two models: 1) a model with continuous $eGFR_{cr}$ as a main predictor, and 2) a model with continuous $eGFR_{cys}$ as a main predictor. We compared receiver-operator characteristic (ROC) curves by treating

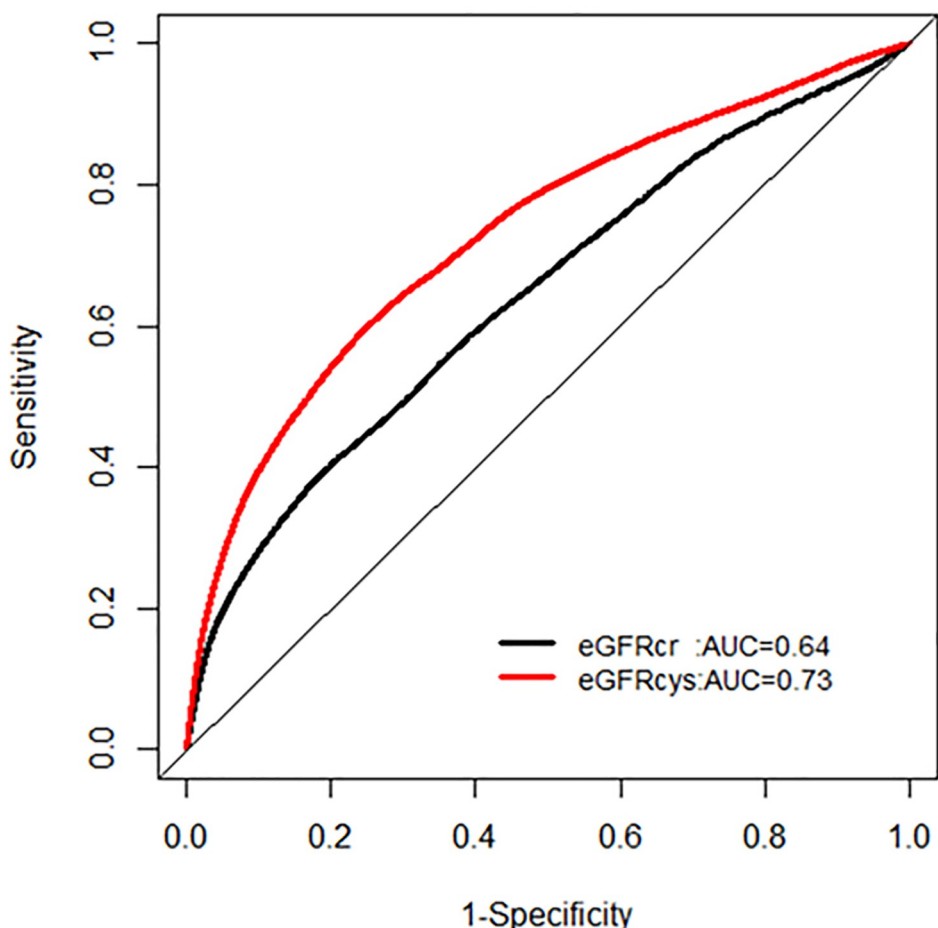

**Fig 5. Comparison of area under the curve for all-cause mortality at 10 years for glomerular filtration rate estimated by serum creatinine (eGFRcr) and cystatin-C (eGFRcys).**

**Table 3. Comparison of the 5 year estimated mortality risk between the model with eGFRcys and the model with eGFRcr.**

| 5-year mortality risk based on Model with eGFRcr | 5-year mortality risk based Model with eGFRcys | | | | | Reclassified |
|---|---|---|---|---|---|---|
| | < 5% | 5%~10% | 10%~20% | > = 20% | Total | n (%) |
| **< 5%** | | | | | | |
| Participants, n (%) | 854 (92.3) | 71(7.7) | 0(0.0) | 0(0.0) | 925 | 71 (7.7) |
| actual event rate (%) | 2.7 | 7.0 | 0.0 | 0.0 | | |
| **5%~10%** | | | | | | |
| Participants, n (%) | 122 (16.8) | 522 (71.9) | 82 (11.3) | 0 (0.0) | 726 | 204 (28.1) |
| actual event rate (%) | 2.5 | 5.0 | 19.5 | 0.0 | | |
| **10%~20%** | | | | | | |
| Participants, n (%) | 1 (0.2) | 93 (14.5) | 482 (75.2) | 65 (10.1) | 641 | 159 (24.8) |
| actual event rate (%) | 0.0 | 5.4 | 13.7 | 44.6 | | |
| **> = 20%** | | | | | | |
| Participants, n (%) | 0 (0.0) | 1 (0.2) | 63 (10.8) | 521 (89.1) | 585 | 64 (10.9) |
| actual event rate (%) | 0.0 | 0.0 | 20.6 | 37.0 | | |

**Table 4. Comparison of the 10 year estimated mortality risk between the model with eGFRcys and the model with eGFRcr.**

| 10 year mortality risk based on Model with eGFRcr | 10 year mortality risk based on Model with eGFRcys | | | | | Reclassified |
|---|---|---|---|---|---|---|
| | < 5% | 5%~10% | 10%~20% | > = 20% | Total | n |
| **< 5%** | | | | | | |
| Participants, n (%) | 174 (87.0) | 26 (13.0) | 0 (0.0) | 0 (0.0) | 200 | 26 (13.0) |
| actual event rate (%) | 2.9 | 0.0 | 0.0 | 0.0 | | |
| **5%~10%** | | | | | | |
| Participants, n (%) | 77 (15.5) | 358 (72.0) | 61 (12.3) | 1 (0.2) | 497 | 139 (28.0) |
| actual event rate (%) | 5.2 | 6.1 | 16.4 | 0.0 | | |
| **10%~20%** | | | | | | |
| Participants, n (%) | 3 (0.4) | 108 (15.4) | 508 (72.6) | 81 (11.6) | 700 | 192 (27.4) |
| actual event rate (%) | 0.0 | 13.9 | 13.0 | 21.0 | | |
| **> = 20%** | | | | | | |
| Participants, n (%) | 0 (0.0) | 1 (0.1) | 96 (6.5) | 1374 (93.4) | 1471 | 97 (6.6) |
| actual event rate (%) | 0.0 | 0.0 | 16.7 | 47.3 | | |
| | | | | | 2868 | 454 |

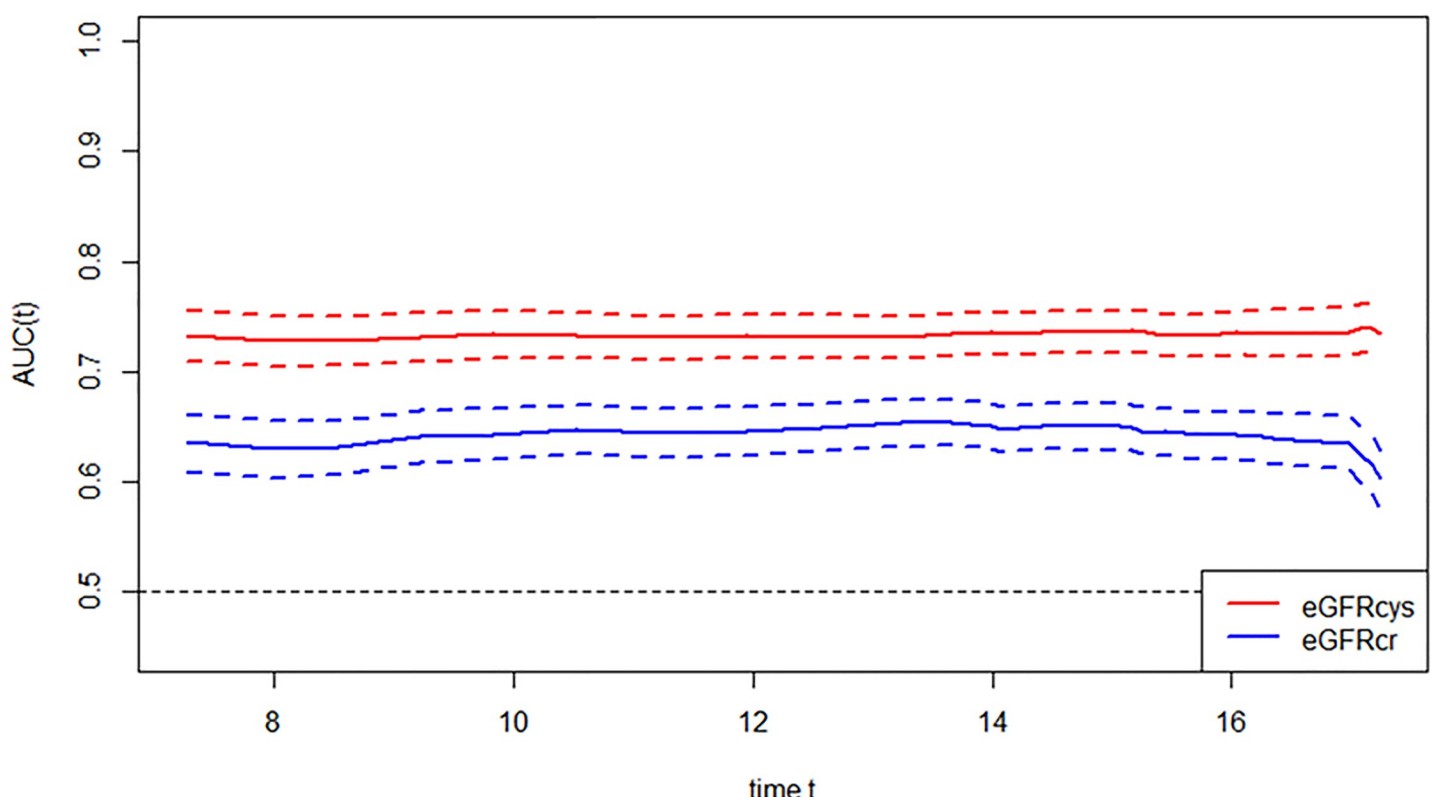

**Fig 6. Area under the curve (AUC) comparisons for model fit for all-cause mortality of glomerular filtration rate estimated by serum creatinine (eGFRcr) and cystatin-C (eGFRcys) stratified by age.**

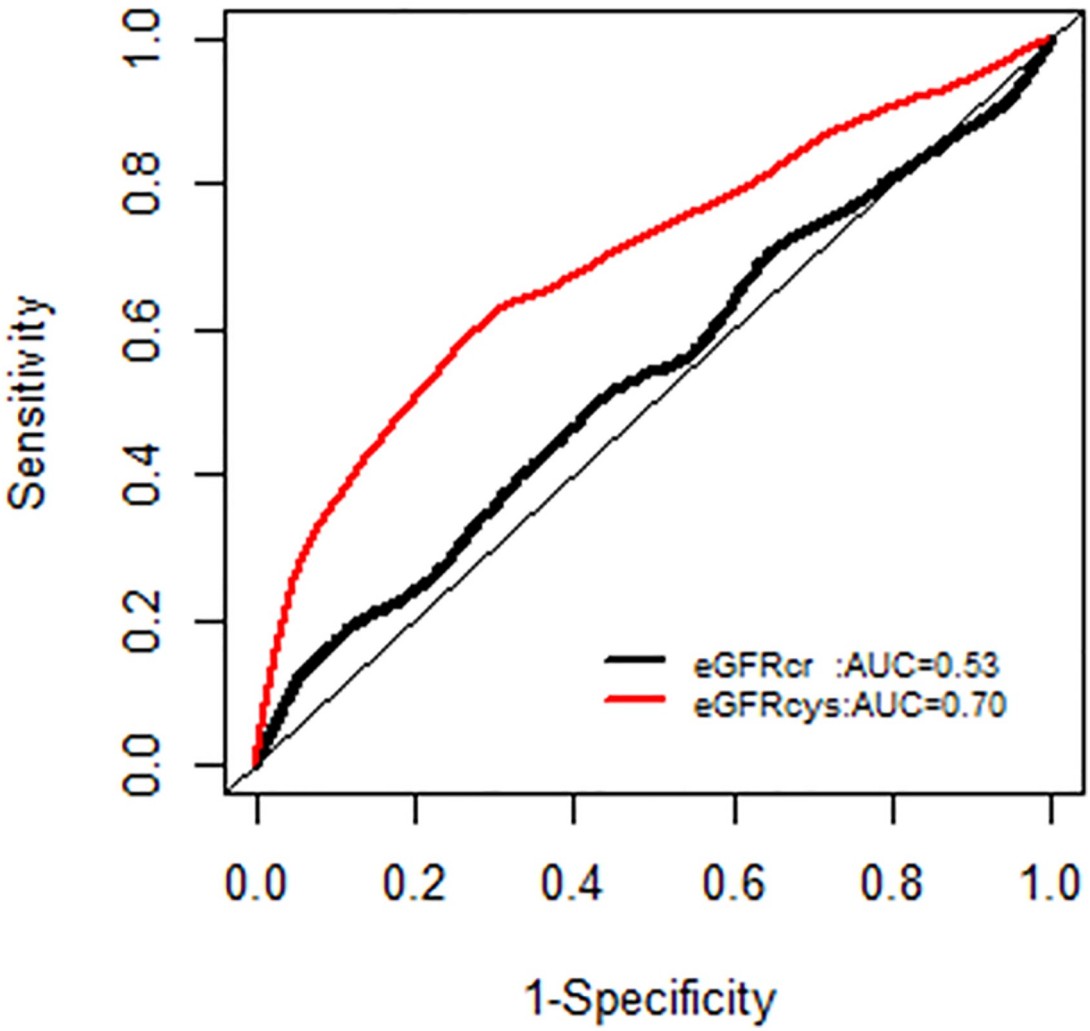

**Fig 7. Area under the curve (AUC) comparisons for model fit for all-cause mortality of glomerular filtration rate estimated by serum creatinine (eGFRcr) and cystatin-C (eGFRcys), age < 70 years at 5 years.**

mortality as a binary outcome, and calculated area under the curve (AUC). To account for censoring, we additionally examined whether AUC changed over time. We also compared the estimated Net Reclassification Improvement (NRI) based on Reynold's 5-year and 10-year mortality risk scores given the high proportion of women in our cohort [22, 23]. Reynold's mortality risk scores were calculated using adjusted Cox proportional hazard models with mortality as an outcome, and then the calculated predicted mortality risk probabilities were categorized as <5%, 5–10%, 10–20% and >20% in order to examine NRI. We further examined the modification effect of NRI by age <70 vs > = 70, sex and race-ethnicity.

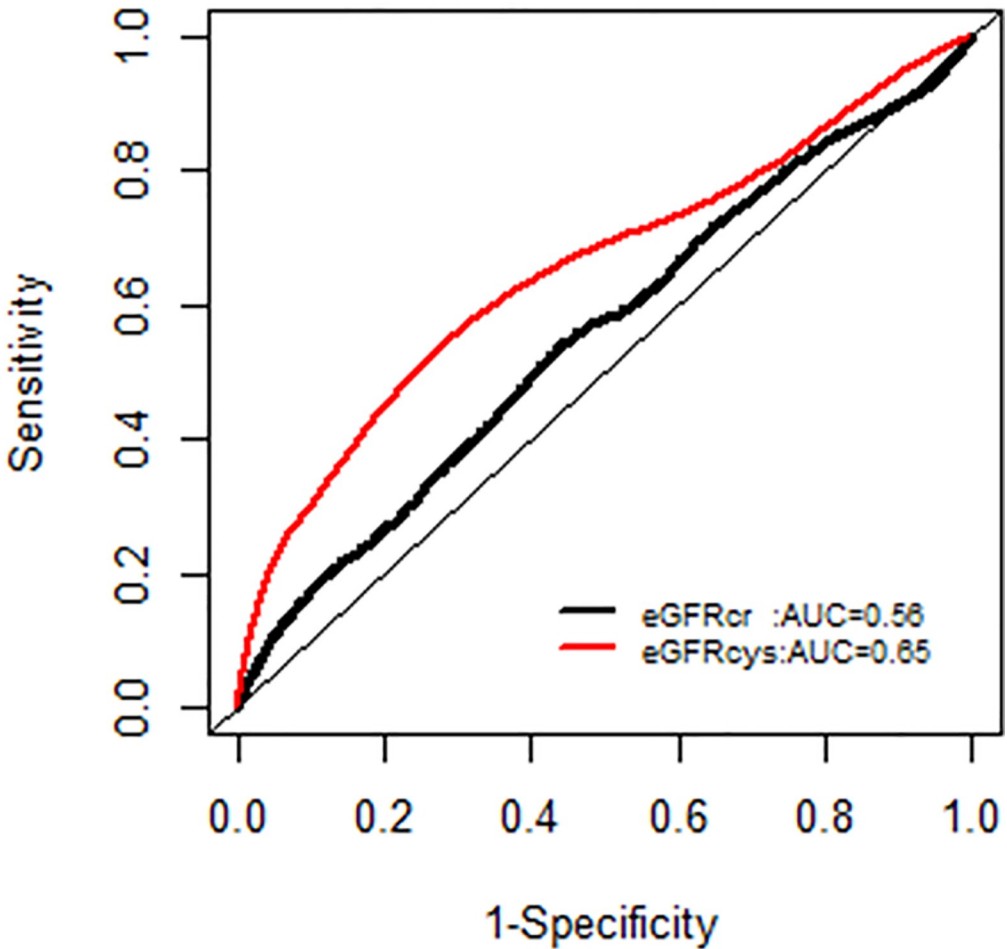

**Fig 8. Area under the curve (AUC) comparisons for model fit for all-cause mortality of glomerular filtration rate estimated by serum creatinine (eGFRcr) and cystatin-C (eGFRcys), age $<$ 70 years at 10 years.**

## Results

There were 2988 participants with both serum creatinine and cystatin-C available. The mean age was 69±10.2 years and participants were predominantly Hispanic (53%) or black (24%), overweight (69%), and current or former smokers (53% combined). The mean eGFRcr (74.68±18.8 ml/min/1.73m$^2$) was higher than eGFRcys (51.72±17.2 ml/min/1.73m$^2$); there was a greater difference in GFR estimations at the upper rather than lower ranges (Figs 1 and 2).

Baseline characteristics are summarized in Table 1. Over a mean of 13 years there were 350 strokes, 208 myocardial infarctions, 475 vascular deaths, 810 non-vascular deaths, and 1611 all-cause deaths (n = 326 unclassified deaths).

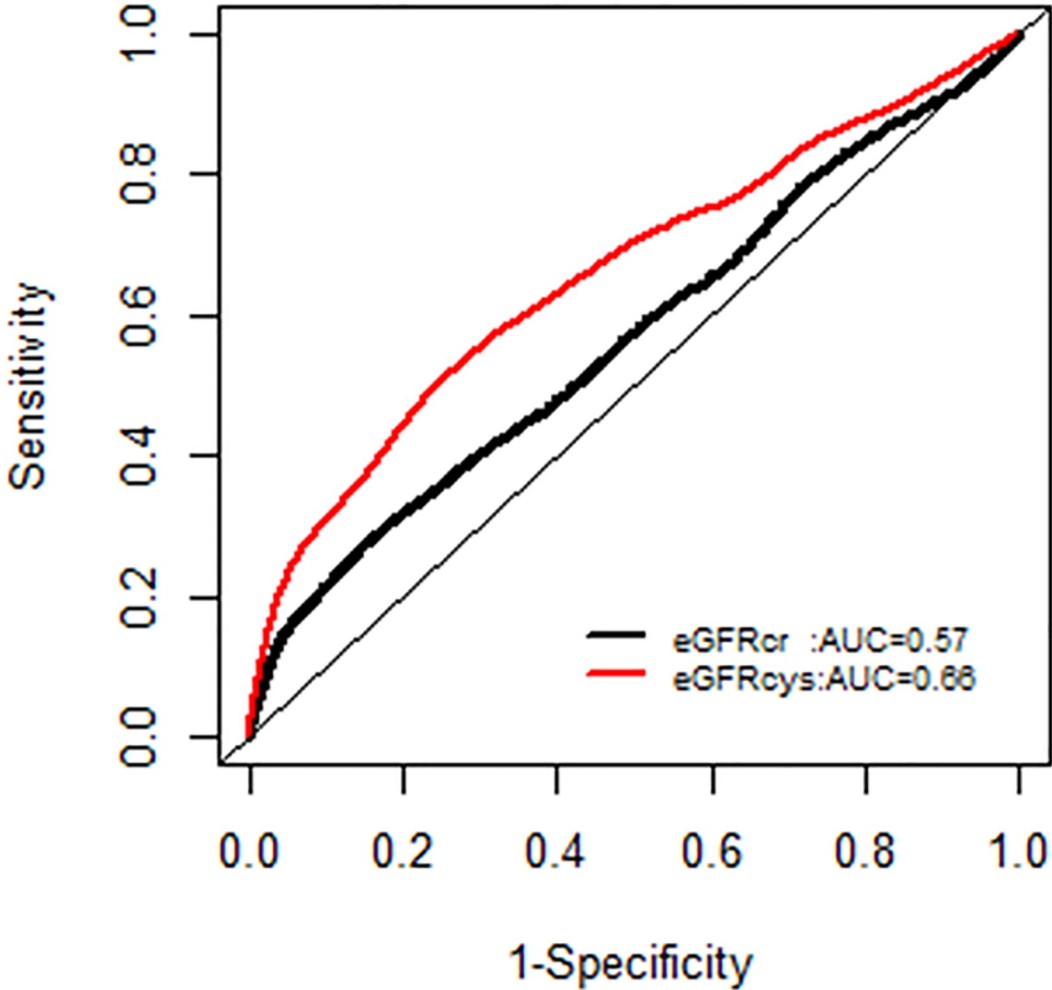

**Fig 9. Area under the curve (AUC) comparisons for model fit for all-cause mortality of glomerular filtration rate estimated by serum creatinine (eGFRcr) and cystatin-C (eGFRcys), age $\geq$ 70 years at 5 years.**

### Association of eGFR with outcomes

In unadjusted models we found that eGFRcr<60 ml/min/1.73m$^2$ and eGFRcys<60 ml/min/1.73m$^2$ were both associated with an increased risk of vascular and non-vascular mortality, as well as the combined vascular endpoint of stroke/MI/vascular death (Table 2). In multi-variable models the associations were somewhat attenuated but remained significant for all-cause mortality for both eGFR$_{cr}$<60 ml/min/1.73m$^2$ (adjusted HR 1.24, 95%CI 1.11–1.39) and eGFR$_{cys}$<60 ml/min/1.73m$^2$ (adjusted HR 1.41, 95% CI 1.22–1.63). eGFRcys<60 ml/min/1.73m$^2$ was associated with vascular and non-vascular mortality; eGFR$_{cr}$ was only associated with non-vascular mortality. Both estimates of GFR< 60 were associated with the combined vascular end-point. The eGFR$_{cr}$< 60 ml/min/1.73m$^2$ (adjusted HR 1.50, 95% CI 1.09–2.06), but not eGFR$_{cys}$< 60 ml/min/1.73m$^2$ (adjusted HR 1.38, 95% CI 0.96–2.00), was

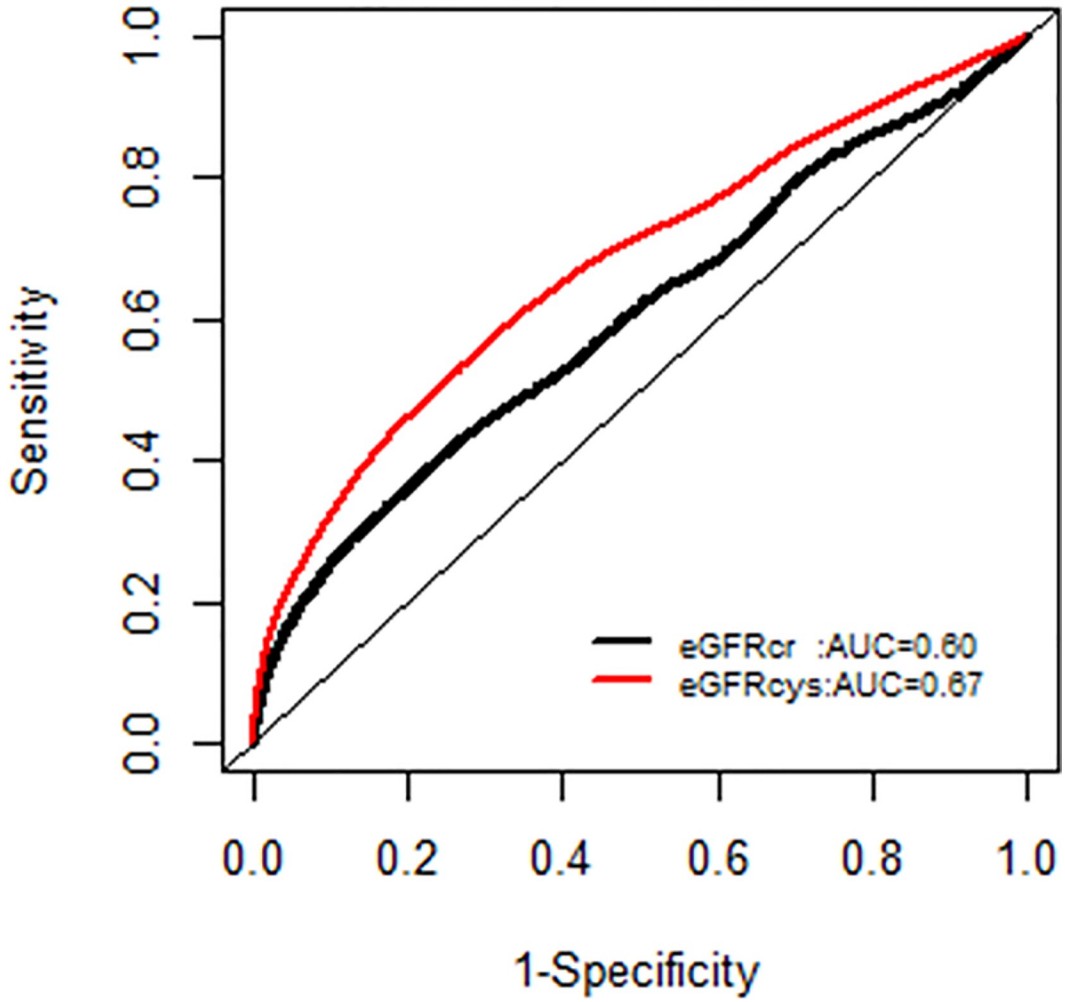

**Fig 10. Area under the curve (AUC) comparisons for model fit for all-cause mortality of glomerular filtration rate estimated by serum creatinine (eGFRcr) and cystatin-C (eGFRcys), age ≥ 70 years at 10 years.**

associated with MI. Neither of the estimates of eGFR< 60 ml/min/1.73m$^2$ was associated with risk of stroke in adjusted models. The results examining eGFR per 10 ml/min/1.73m$^2$ increments were similar to the categorical definitions for eGFR except for eGFR$_{cr}$ no longer being associated with the combined end-point, and eGFR$_{cys}$ being associated with the risk of MI (Table 2).

## Comparison in mortality risk predictions

In order to examine the performance in mortality risk prediction, we first compared the model fit using ROC curves (Fig 3).

# Women at Year = 5

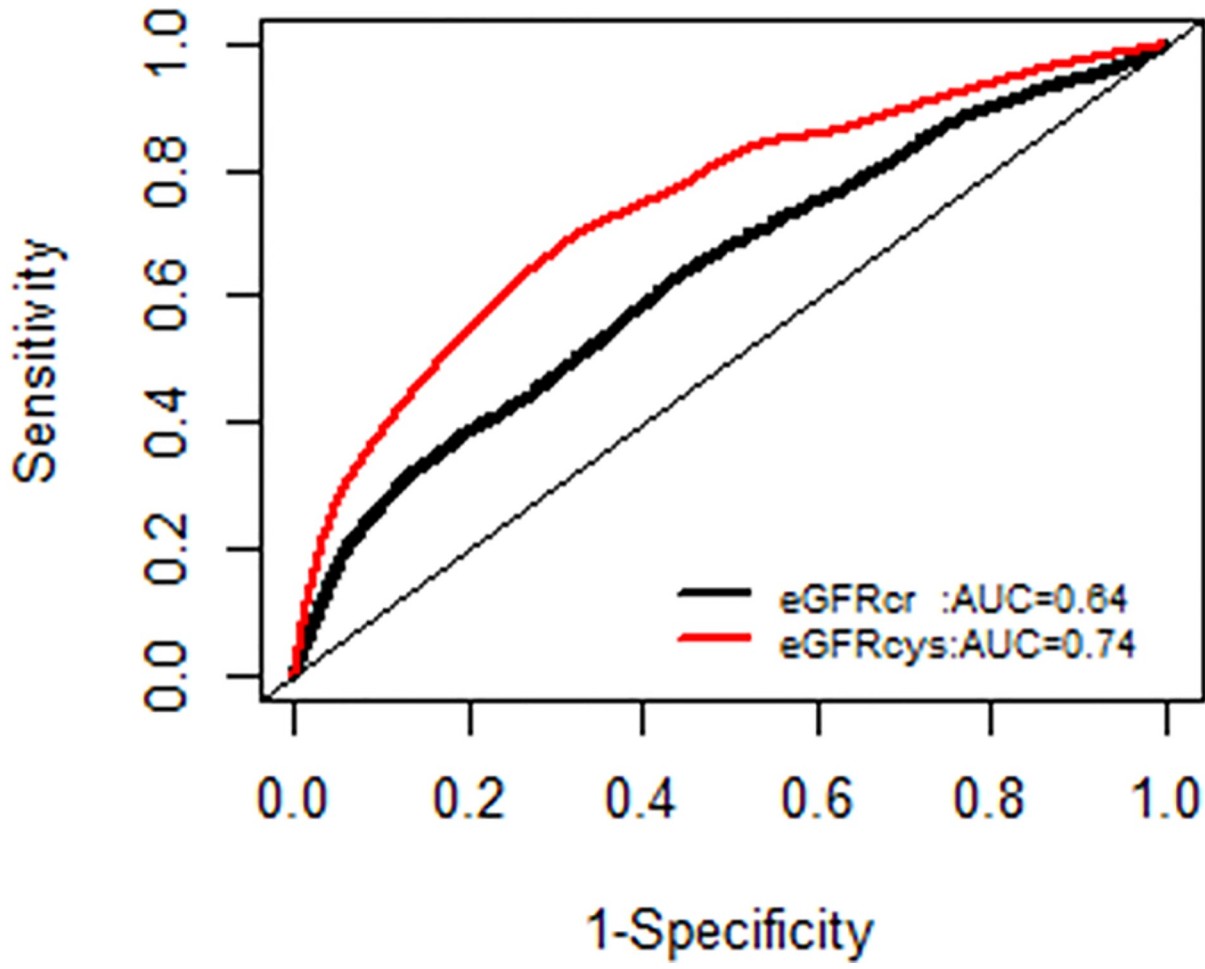

**Fig 11. Area under the curve (AUC) comparisons for model fit for all-cause mortality of glomerular filtration rate estimated by serum creatinine (eGFRcr) and cystatin-C (eGFRcys), women at 5 years.**

We found that eGFR$_{cys}$ was associated with an improved model performance compared to eGFR$_{cr}$ (AUC 0.73 vs 0.67, p for difference< 0.0001) when mortality was treated as a binary outcome.

We further examined whether the AUC for models with each eGFR was changing over the years of follow-up, and found no significant change in AUC over time (Figs 4 and 5).

The proportion of correct reclassification by the model with eGFR$_{cys}$ compared to the model with eGFR$_{cr}$ was 4% based on Reynold's 10-year risk (p = 0.002) and 11% on 5-year risk (p < .0001), respectively (Tables 3 and 4).

When the interactions of NRI with age<70 vs. > = 70 were examined, there was a significant difference in NRI based on 5 year mortality risk; the proportion of correct reclassification by the model with eGFR$_{cys}$ compared to eGFR$_{cr}$ was greater among those with age <70 than

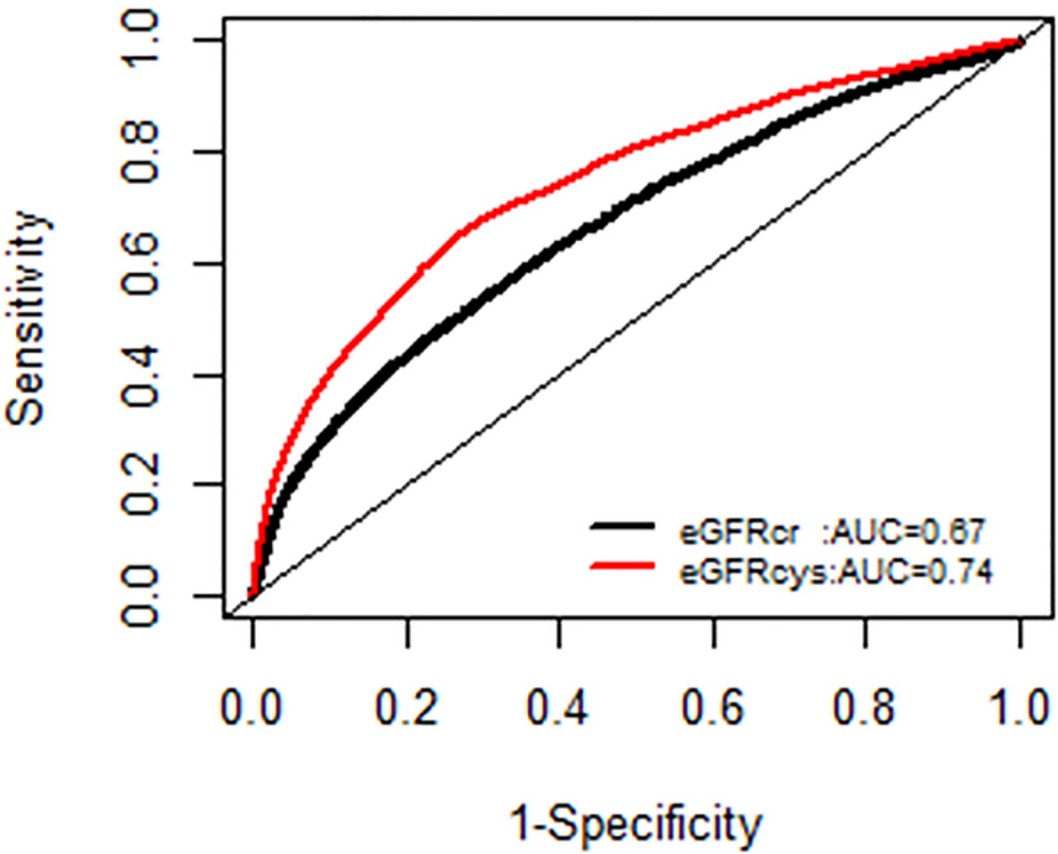

**Fig 12. Area under the curve (AUC) comparisons for model fit for all-cause mortality of glomerular filtration rate estimated by serum creatinine (eGFRcr) and cystatin-C (eGFRcys), women at 10 years.**

age> = 70 (estimated NRI = 22% for age<70 group and 9% for age> = 70 group: p for difference = 0.047) We also found an interaction by sex such that the proportion of correct reclassification was higher in men than in women (estimated NRI for women 7%, men 17%, p for difference = 0.049) with eGFR$_{cys}$ compared to eGFR$_{cr}$.

The AUC for models with each GFR supported the improved model fit in this age group (Fig 6).

We, however, found no interactions of NRI based on 10 year mortality risk by age groups. Similarly there were no modification effects of NRI by sex or race-ethnicity, and the AUC for each GFR equation by race-ethnicity was similar (Figs 7–20).

## Discussion

In an elderly race/ethnically diverse cohort with a large burden of hypertension and diabetes we found that CKD defined by either serum creatinine or cystatin-C based eGFR was associated with an increased risk of mortality, especially vascular death. However, no significant

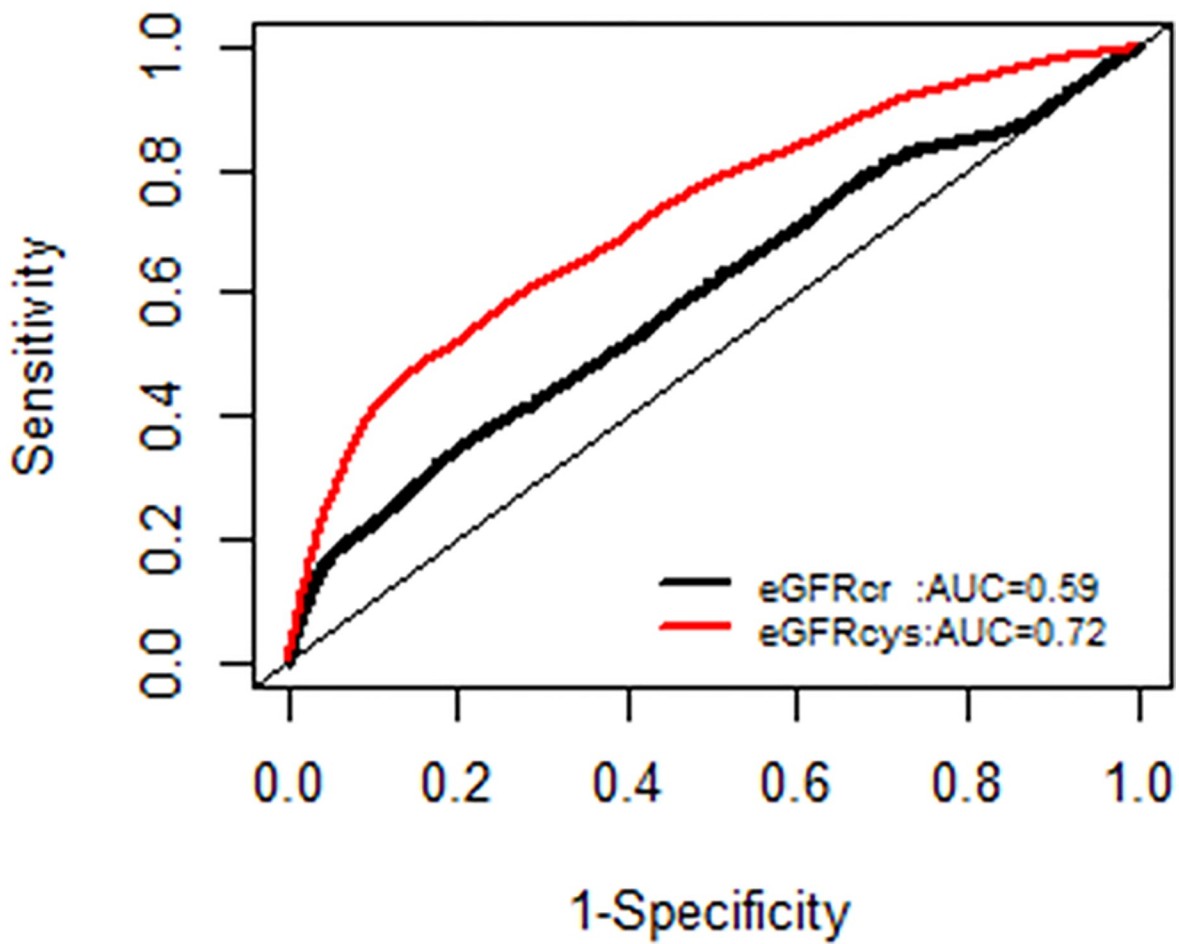

**Fig 13. Area under the curve (AUC) comparisons for model fit for all-cause mortality of glomerular filtration rate estimated by serum creatinine (eGFRcr) and cystatin-C (eGFRcys), men at 5 years.**

associations with non-fatal CVD events such as stroke or myocardial infarction were found. Though both estimates of eGFR were associated with the risk of death, the estimated GFR using serum cystatin-C was better in predicting 5-year and 10-year mortality risk. Notably, eGFR$_{cys}$ outperformed eGFR$_{cr}$ in predicting 5-year mortality risk especially among those with age <70 years; this same age group was more likely to be black and Hispanic compared to white.

Our results, in an elderly multiethnic cohort, are in keeping with the established association of CKD with mortality, particularly with cystatin-C based estimates in the Cardiovascular Health Study for example [24]. The results related to eGFR$_{cys}$ are consistent with findings from other studies suggesting that eGFR$_{cys}$ may be a more accurate estimate of GFR than a serum creatinine-based formula [25], and extend those findings to an elderly multiethnic population where GFR$_{cr}$ may be confounded by loss of muscle mass which would attenuate the

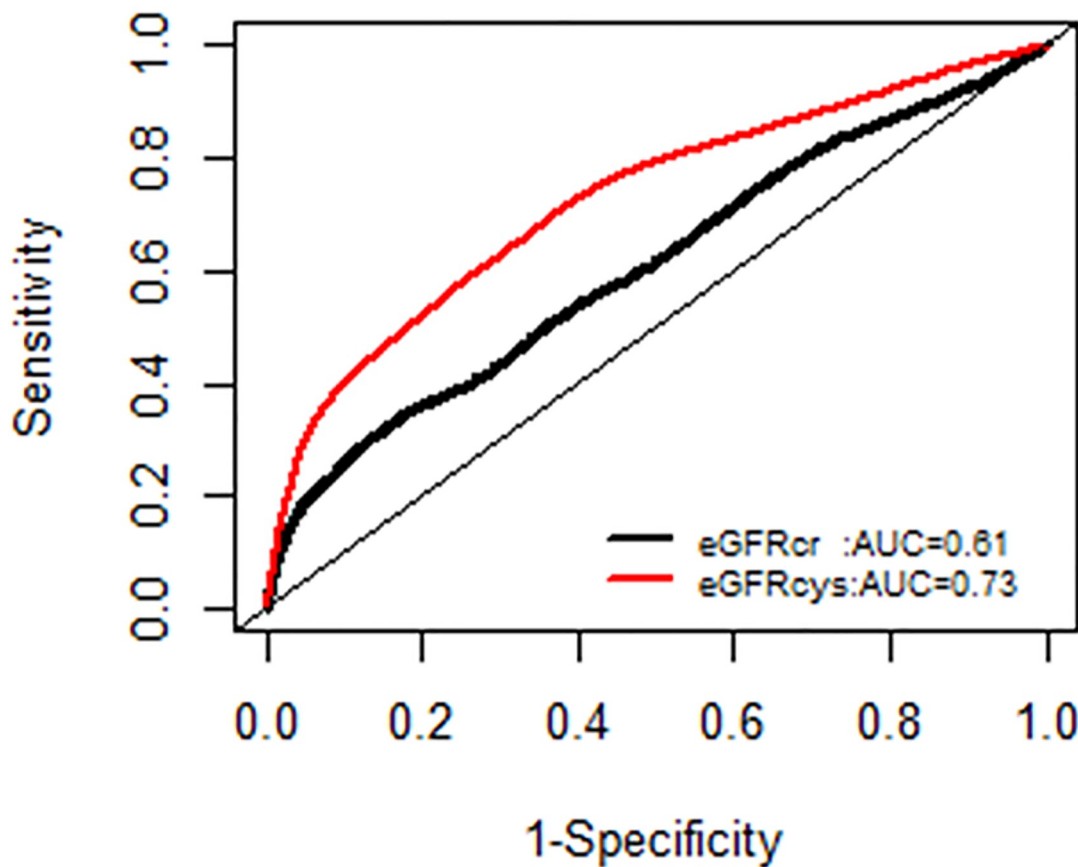

**Fig 14. Area under the curve (AUC) comparisons for model fit for all-cause mortality of glomerular filtration rate estimated by serum creatinine (eGFRcr) and cystatin-C (eGFRcys), men at 10 years.**

association. The inability to accurately estimate GFR disproportionately affects women, blacks and Hispanic elderly patients creating significant challenges for prognostication for outcomes, decline of renal function, and management (particularly for medication dosing) of these individuals. For example, in the NOMAS cohort, creatinine and cystatin based eGFR have resulted in dramatically different estimates of CKD prevalence (21.9% and 70.5% respectively) calling into question the precision of the eGFR equations. Similar divergent results have been reported by other albeit smaller cohorts [26–28]. Interestingly in our cohort the predictive ability of eGFR (regardless of serum measure) appeared higher in the younger participants and men who were most likely to be included in prior cohort that derived GFR estimation formulae. These results highlight the importance of improved accuracy in measurement of GFR in diverse populations will help better understand how CKD is associated with CVD mortality related disparities.

The most widely accepted equations to estimate GFR using serum creatinine in adults include the Modification of Diet in Renal Disease (MDRD) and the more recent CKD Epidemiology Collaboration (CKD-EPI) equation. The latter, which includes the same variables as

## white

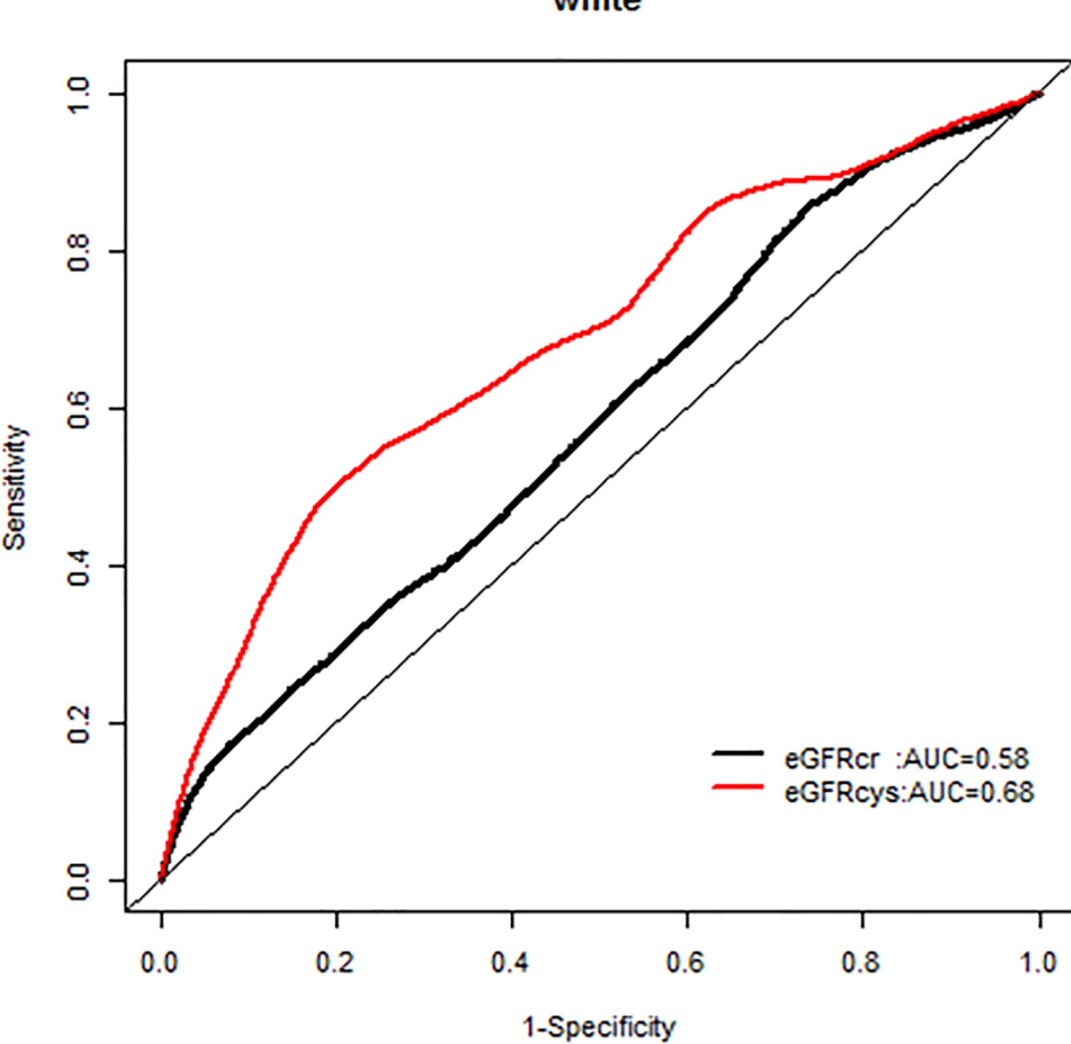

**Fig 15. Area under the curve (AUC) comparisons for model fit for all-cause mortality of glomerular filtration rate estimated by serum creatinine (eGFRcr) and cystatin-C (eGFRcys), whites at 5 years.**

the MDRD equation but with different coefficients, has been described as a more accurate estimate of GFR across the range of renal function, especially for eGFR > 60mL/min, and provides better risk stratification in the general population [29, 30]. However, creatinine generation is directly related to muscle mass, and creatinine based eGFR estimation is therefore impacted by age and other circumstances that result in a change in body composition such as sarcopenia [31], limb loss, as well as functional impairments [32]. The estimation of GFR using serum cystatin-C, an endogenous protease inhibitor produced by all nucleated cells and filtered freely by the kidneys, has been proposed as a potentially more accurate method of renal function assessment and a better prognostication marker, particularly in the elderly and diverse populations [24, 33–36]. Serum cystatin-C based GFR

## Black

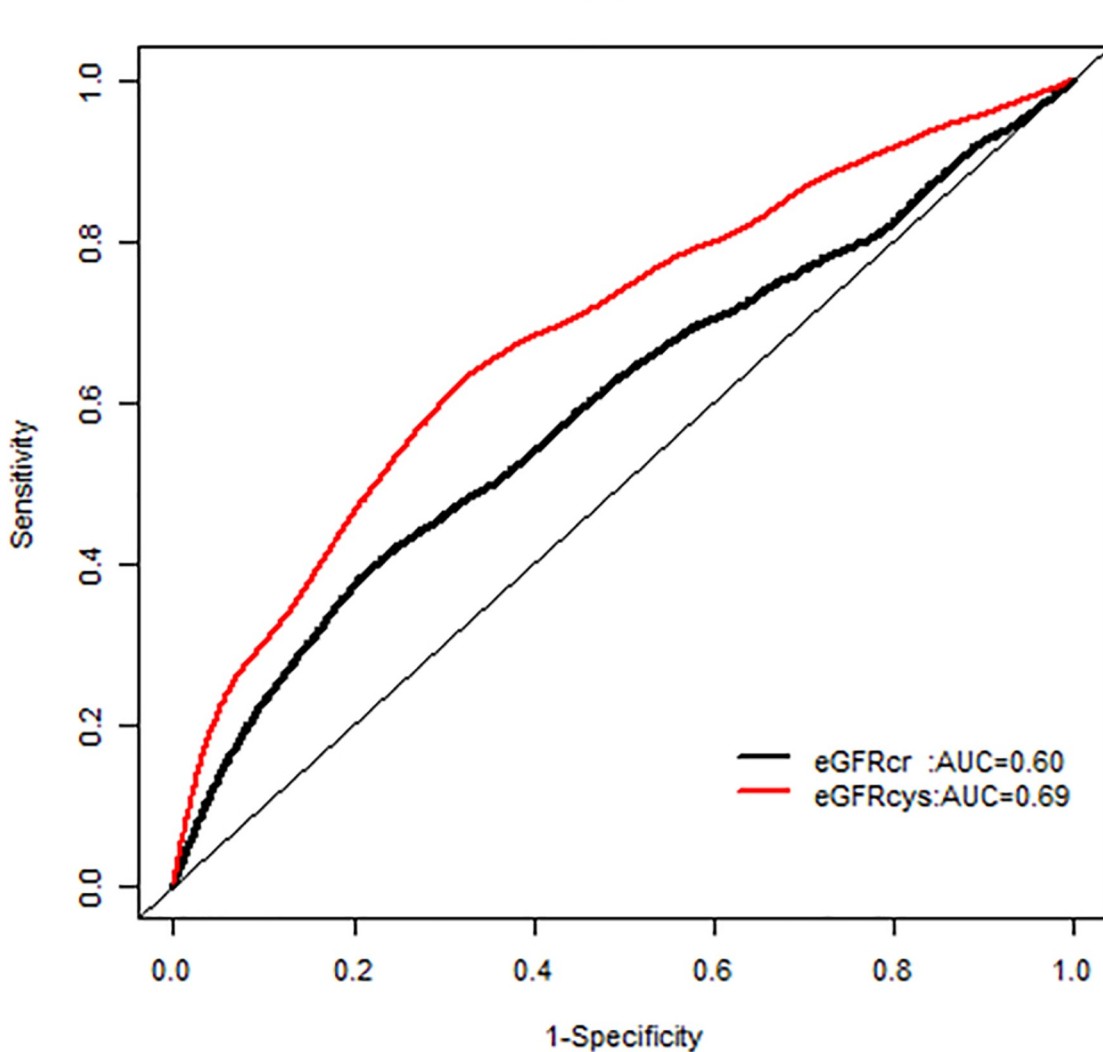

**Fig 16. Area under the curve (AUC) comparisons for model fit for all-cause mortality of glomerular filtration rate estimated by serum creatinine (eGFRcr) and cystatin-C (eGFRcys), blacks at 5 years.**

estimation has not yet been widely adopted in clinical practice and more recently has been recognized to also be affected by aging, raising questions about the interpretation of GFR estimates in the elderly [37].

A further significant concern regarding currently used eGFR formulas is their generalizability to populations with substantial proportions of elderly and Hispanics. For example the Cockcroft-Gault formula was initially derived in 1979 in a study of only white men [38]. The MDRD equation was developed in a cohort of 1628 participants with a mean age of 50 ± 13 years, 60% men, and 88% non-Hispanic white [39]. The CKD-EPI equation was developed in 5504 participants including only <5% of the sample over age 70, though race-ethnicity representation was slightly improved (32% black, 5% Hispanic) [20]. Additional formulas have been

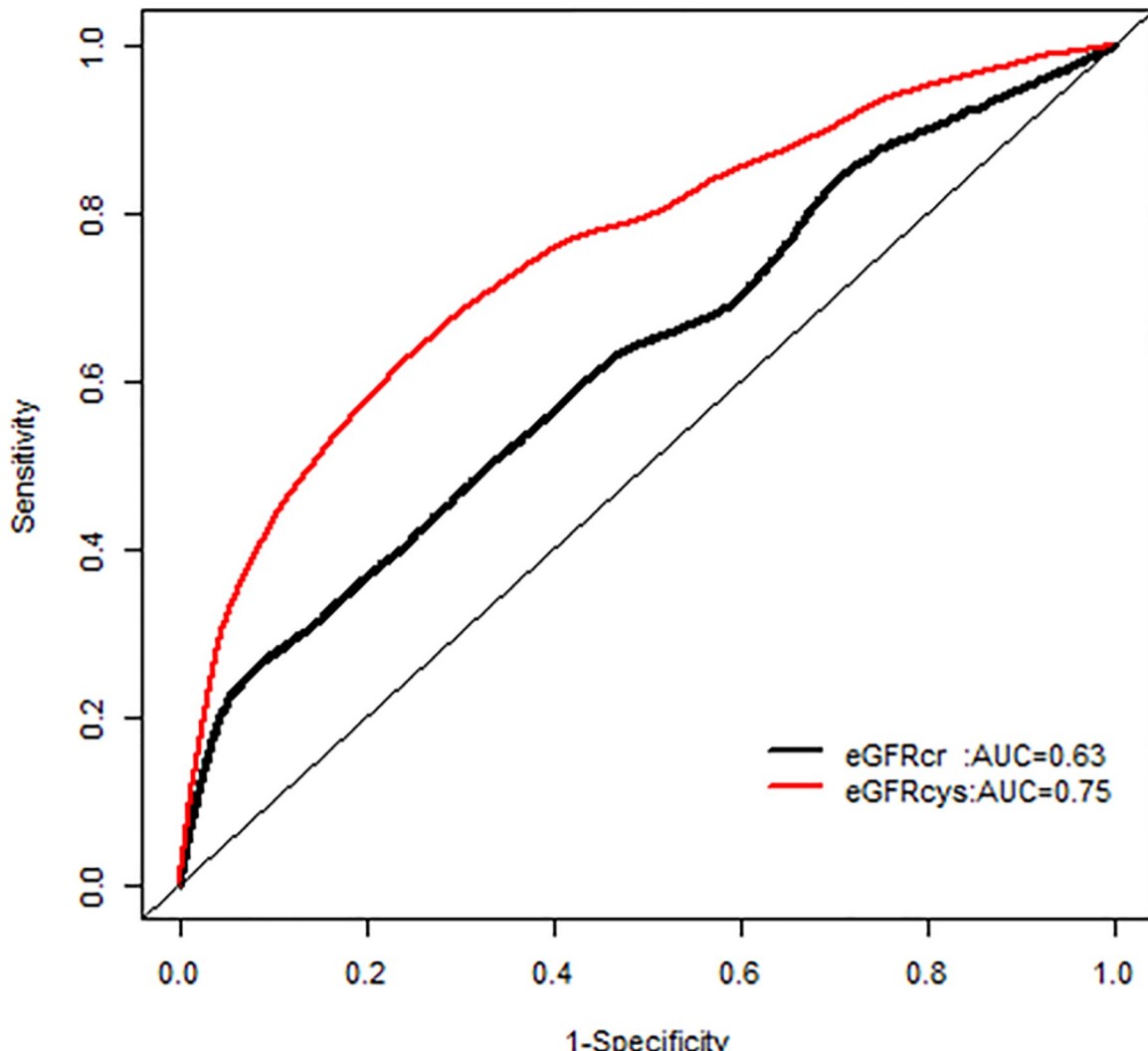

**Fig 17. Area under the curve (AUC) comparisons for model fit for all-cause mortality of glomerular filtration rate estimated by serum creatinine (eGFRcr) and cystatin-C (eGFRcys), Hispanics at 5 years.**

proposed using both serum creatinine and cystatin-C with even lower proportions of elderly blacks and Hispanics [25, 40]. Newer estimates, including the Chronic Renal Insufficiency Cohort GFR estimating equation, performed poorly among Hispanics, blacks, and elderly [41]. Similarly, the Berlin Initiative Study (BIS) using both serum creatinine and cystatin-C (mean age 78.5) in a European population did not perform well in accurately predicting renal function in other populations with higher proportions non-whites [28]. Unfortunately, there is a paucity of data on the most accurate GFR formula to use in diverse populations despite prior research documenting differences in serum creatinine and cystatin-C by age and race-ethnicity

## white

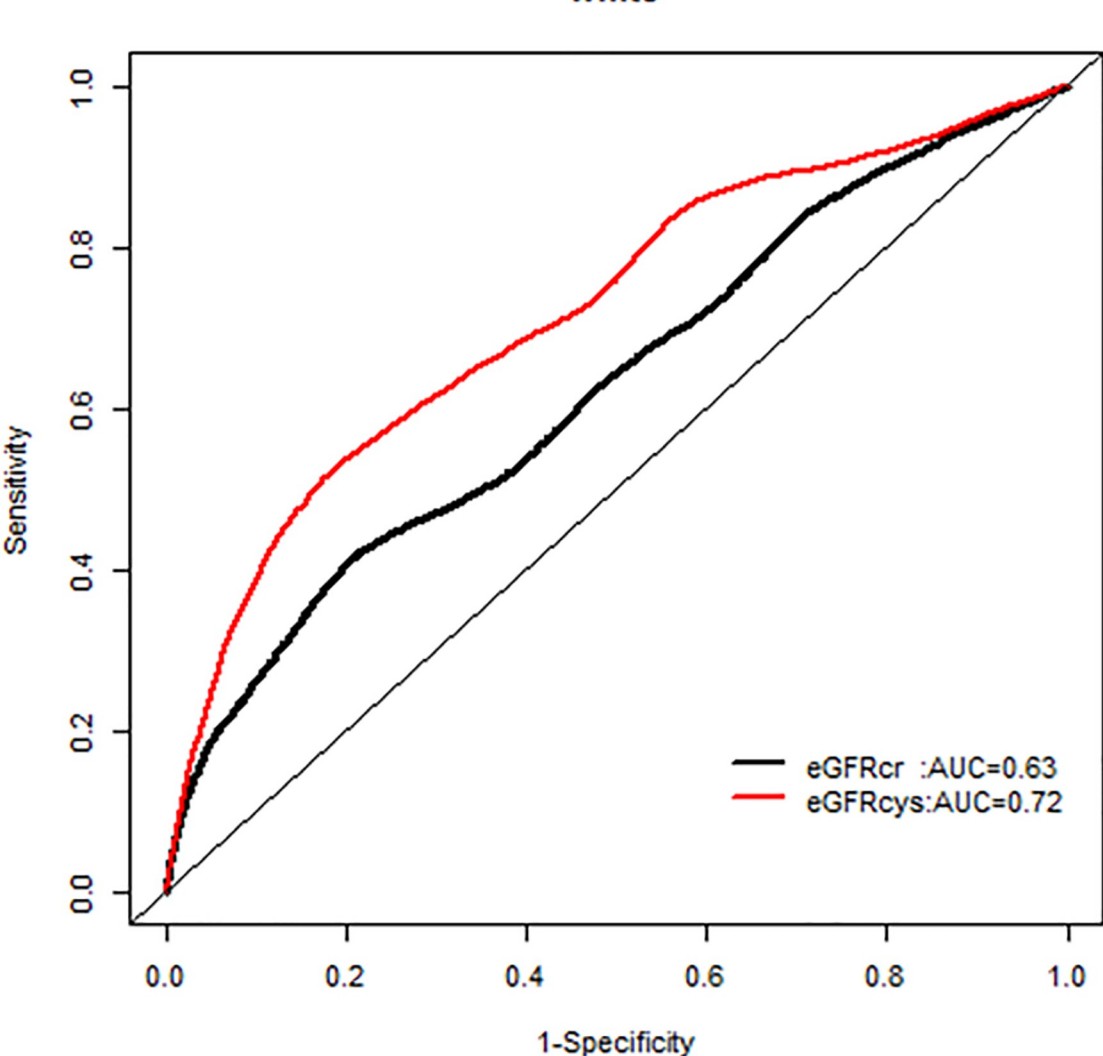

eGFRcr  :AUC=0.63
eGFRcys:AUC=0.72

**Fig 18. Area under the curve (AUC) comparisons for model fit for all-cause mortality of glomerular filtration rate estimated by serum creatinine (eGFRcr) and cystatin-C (eGFRcys), whites at 10 years.**

[42]. Population-based studies in the United States with large proportions of diverse participants have been limited to smaller sample sizes such as the 294 participants in MESA-Kidney [43].

The strengths of our study include a large multi-ethnic population and comprehensive follow up for death and CVD events over 10 years. Our study has several weaknesses that require discussion. In NOMAS, we did not measure GFR with an exogenous marker such as iohexol or iothalamate and as such cannot determine which serum marker provides the most accurate estimate of the measured GFR in absolute terms. Instead, we focused on predictive validity, which may have other clinical advantages beyond the mere estimation of renal function.

**Black**

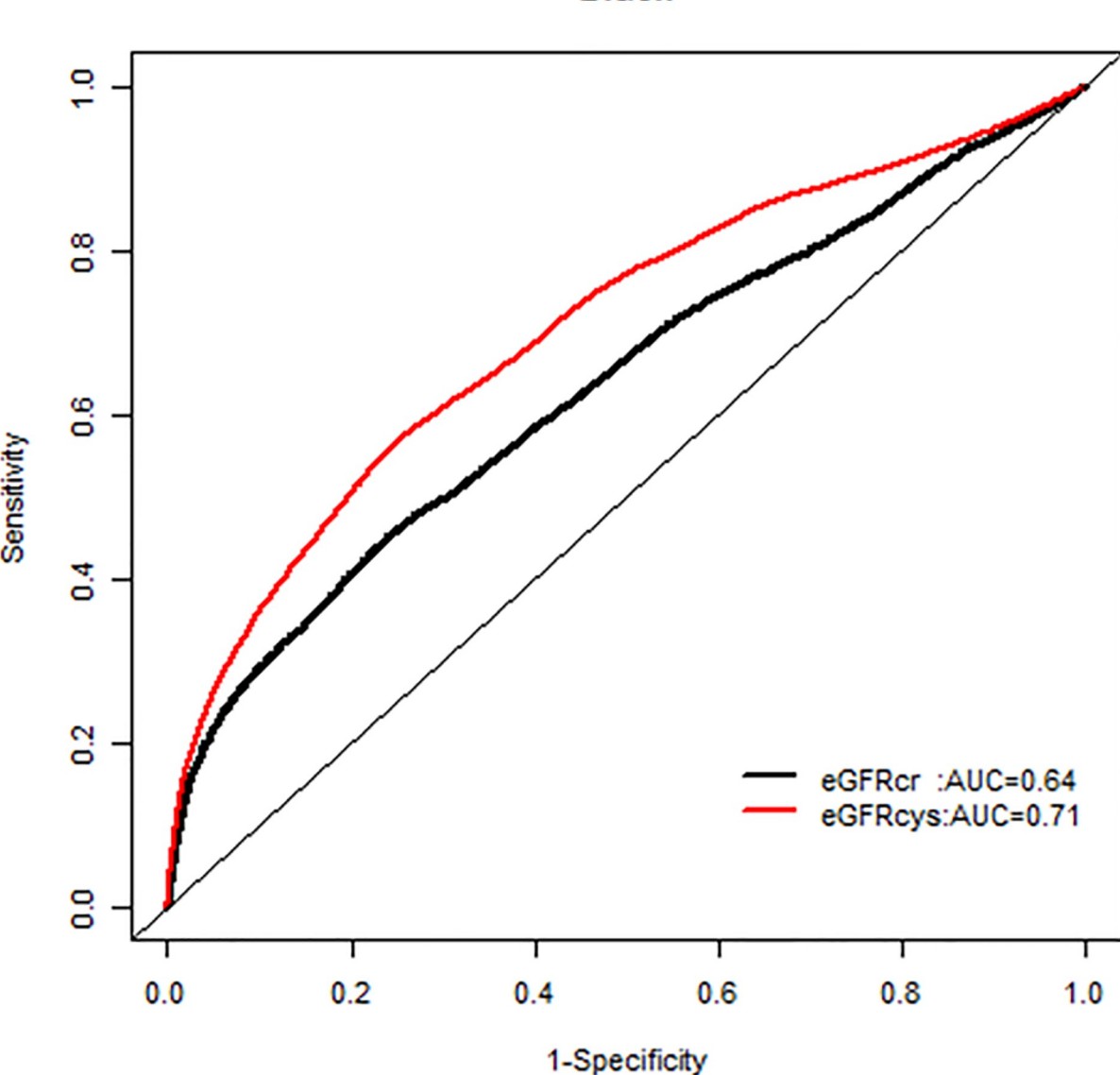

**Fig 19. Area under the curve (AUC) comparisons for model fit for all-cause mortality of glomerular filtration rate estimated by serum creatinine (eGFRcr) and cystatin-C (eGFRcys), blacks at 10 years.**

NOMAS did not obtain repeated measures of serum creatinine or cystatin-C to document a decline in values over time, nor did we systematically determine whether participants progressed to end stage renal disease. This data would provide additional information on which marker better predicted prevalent higher stages of CKD. Cystatin-C levels can be affected by several medical conditions including thyroid dysfunction [44] and human immunodeficiency virus infection [45] which unfortunately we did not collect in NOMAS. Lastly, we did not collect urine protein measurements that would help identify CKD in patients with eGFR >60mL/ min or improve the risk prediction models.

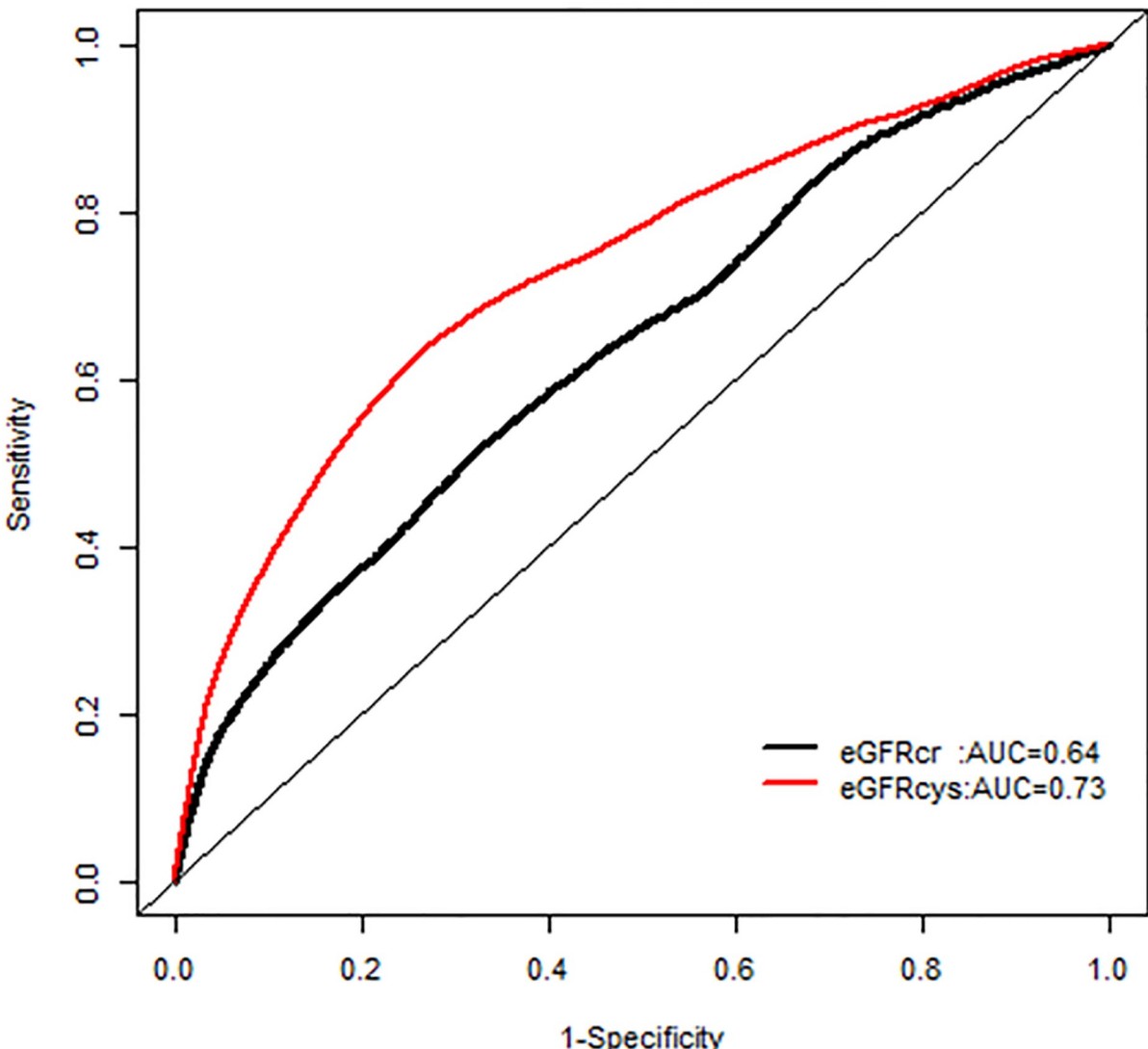

**Fig 20. Area under the curve (AUC) comparisons for model fit for all-cause mortality of glomerular filtration rate estimated by serum creatinine (eGFRcr) and cystatin-C (eGFRcys), Hispanics at 10 years.**

In conclusion, eGFRcys provided a better prognostication tool for the risk of mortality compared to eGFRcr. Further research is required in diverse populations, including elderly multi-ethnic populations, on accurately measuring GFR.

## Author Contributions

**Conceptualization:** Joshua Z. Willey, Clinton B. Wright.

**Formal analysis:** Yeseon Park Moon.

**Methodology:** S. Ali Husain, Myles Wolf, Sumit Mohan.

**Supervision:** Mitchell S. V. Elkind, Ralph L. Sacco, Ken Cheung, Clinton B. Wright, Sumit Mohan.

**Writing – original draft:** Joshua Z. Willey.

**Writing – review & editing:** Yeseon Park Moon, S. Ali Husain, Mitchell S. V. Elkind, Ralph L. Sacco, Myles Wolf, Ken Cheung, Clinton B. Wright, Sumit Mohan.

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
