## [Decision Letter · Decision Letter 0]

19 Aug 2019

PONE-D-19-20793

Creatinine versus Cystatin C for Renal Function-Based Mortality Prediction in an Elderly Cohort: the Northern Manhattan Study

PLOS ONE

Dear Dr Willey,

Thank you for submitting your manuscript to PLOS ONE. After careful consideration, we feel that it has merit but does not fully meet PLOS ONE’s publication criteria as it currently stands. Therefore, we invite you to submit a revised version of the manuscript that addresses the points raised during the review process.

We would appreciate receiving your revised manuscript by Oct 03 2019 11:59PM. To enhance the reproducibility of your results, we recommend that if applicable you deposit your laboratory protocols in protocols.io, where a protocol can be assigned its own identifier (DOI) such that it can be cited independently in the future. For instructions see: http://journals.plos.org/plosone/s/submission-guidelines#loc-laboratory-protocols

We look forward to receiving your revised manuscript.

Kind regards,

Tatsuo Shimosawa, M.D., Ph.D.

Academic Editor

PLOS ONE

**Journal Requirements:**

**Comments to the Author**

1. Is the manuscript technically sound, and do the data support the conclusions?

Reviewer #1: Yes

Reviewer #2: Yes

2. Has the statistical analysis been performed appropriately and rigorously? 

Reviewer #1: I Don't Know

Reviewer #2: Yes

3. Have the authors made all data underlying the findings in their manuscript fully available?

Reviewer #1: Yes

Reviewer #2: Yes

4. Is the manuscript presented in an intelligible fashion and written in standard English?

Reviewer #1: Yes

Reviewer #2: Yes

5. Review Comments to the Author

Reviewer #1: The manuscript by Willey et al. investigated the role of estimated GFR based on creatinine (eGFRcr) or cystatin-C (eGFRcys) for mortality risk prediction in an elderly, ethnically diverse cohort, and found that eGFRcys was superior in predicting the risk of all-cause mortality. The topic may be of scientific interest, and might give some impact on clinical practice. I feel, however, there are still some unclear and unconvincing points that should be clarified from a scientific viewpoint. My major concerns are as follows:

1. Although there was a slight but significant difference in a predicting power between the two measures, the reason is not clear. As the authors state in Discussion, the most important point must be whether eGFRcys is superior in accurately measuring GFR in their cohort. This point should be clarified at least in a small number of subjects in their cohort, using iohexol or inulin as an exogenous marker.

2. Also, the dot plot between eGFRcr (mean, 74.7) and eGFRcys (mean, 51.7) should be presented to know the correlation between the two. As the CKD stage progresses, does the discrepancy become larger?

3. It is also important and should be analyzed whether the predicting power of eGFRcys was influenced by the severity of CKD, i.e., CKD stage 1-2 (eGFR > 60), stage 3, or stage 4-5 (eGFR < 30). Mortality incidence should have been more often in advanced CKD stages. This point should be clearly presented.

4. In Figure 3, the ROC curve of eGFRcys seemed much better than eGFRcr among the subjects under 70. Why?

5. It is not clear what are critical conditions where eGFRcys is superior to eGFRcr in mortality risk prediction. Males? Younger age? Caucasians? Subgroup analysis should be performed in order to specify the factors which have impact in favor of eGFRcys for risk prediction.

Reviewer #2: The authors compared creatinine and cystatin C as factor of eGFR. I think that the aim of this study is very interesting to nephrologist, and felt old.

1. I believe that authors knew serum cystatin C concentration is influenced by many factors, hyper/hypo thyroid, HIV infection, and so on. I could not find in the manuscript about exclusion of these patient from cohort.

2. Main finding of this study may be “eGFRcys predicted all-cause mortality better than eGFRcre”. However, this fact is already reported indirectly as the authors cited in Ref 24, 33,34, and Astor BC et al 2011, Peralta CA et al 2011

6. PLOS authors have the option to publish the peer review history of their article (what does this mean?). If published, this will include your full peer review and any attached files.

Reviewer #1: No

Reviewer #2: No

---

## [Author Response · Author response to Decision Letter 0]

5 Nov 2019

We appreciate the reviewer’s comments on our manuscript. We have included the comments by the reviewers below and have modified the manuscript as requested in the appropriate sections and included below are responses to the reviews.

Reviewer #1: 

1. Although there was a slight but significant difference in a predicting power between the two measures, the reason is not clear. As the authors state in Discussion, the most important point must be whether eGFRcys is superior in accurately measuring GFR in their cohort. This point should be clarified at least in a small number of subjects in their cohort, using iohexol or inulin as an exogenous marker.

Response:

We agree with the reviewer that having a gold-standard measure of GFR would be ideal. Unfortunately the serum measures were collected at the time of initial enrollment in the Northern Manhattan Study between 1993 to 2001 such that we do not have the ability to measure GFR concomitantly. We currently do not have funding to measure GFR objectively in NOMAS but this is a planned future study if funded. We have acknowledged this is a limitation of our study.

2. Also, the dot plot between eGFRcr (mean, 74.7) and eGFRcys (mean, 51.7) should be presented to know the correlation between the two. As the CKD stage progresses, does the discrepancy become larger?

Response:

We agree and have included the following figure to outline the raw data with a dot and Bland&Altman plot. From our data it seems the discrepancy between eGFRCys and eGFRCr is higher at the higher ranges of GFR estimation than in the lower end. We have included this as the new figure 1 and added a comment in the results section.

Dot plot (left) and the Bland & Altman plot (right) (figure 1)

In results section: “The mean eGFRcr (74.68±18.8 ml/min/1.73m2) was higher than eGFRcys (51.72±17.2 ml/min/1.73m2); there was a greater difference in GFR estimations at the upper rather than lower ranges (figure 1).”

3. It is also important and should be analyzed whether the predicting power of eGFRcys was influenced by the severity of CKD, i.e., CKD stage 1-2 (eGFR > 60), stage 3, or stage 4-5 (eGFR < 30). Mortality incidence should have been more often in advanced CKD stages. This point should be clearly presented.

Response: 

We agree with the reviewer than analyzing by CKD stages would have been ideal in our analyses. Unfortunately the proportion of participants with stages 4-5 in our cohort was small and chose to collapse stages 3-5 together. We were nonetheless concerned that severity of CKD would be important and performed our analyses using GFR in a continuous manner (per 10 ml/min/1.73m2) and noted there was a significant association with all-cause mortality.

4. In Figure 3, the ROC curve of eGFRcys seemed much better than eGFRcr among the subjects under 70. Why?

Response:

We thank the reviewer for highlighting this finding in our study. We were concerned that in the older participants in our study serum creatinine would be less predictive due to loss of muscle mass in the elderly, as well as less validated GFR estimation formulae for multi-ethnic populations such as ours. 

The discussion has been modified as follows:

“The results, particularly in the participants older than age 70, related to eGFRcys are consistent with findings from other studies suggesting that eGFRcys may be a more accurate estimate of GFR than a serum creatinine-based formula, and extend those findings to an elderly multiethnic population where GFRcr may be confounded by loss of muscle mass which would attenuate the association. The inability to accurately estimate GFR disproportionately affects blacks and Hispanic elderly patients creating significant challenges for prognostication for outcomes, decline of renal function, and management (particularly for medication dosing) of these individuals.”

5. It is not clear what are critical conditions where eGFRcys is superior to eGFRcr in mortality risk prediction. Males? Younger age? Caucasians? Subgroup analysis should be performed in order to specify the factors which have impact in favor of eGFRcys for risk prediction.

Response:

We performed analyses examining GFR estimates by sex, age, and race-ethnicity and noted that for predicting 5 year mortality risk, eGFRcys was better than eGFRcr among those age under 70 years old (p for difference=0.047, compared to age>70) or men (p for difference=0.049, compared to woman). No race-ethnicity differences were found. We have included this in the results section.

For predicting 10 year mortality risk, there were no statistically interactions. 

We have included the following table for reference for the reviewer but not in the manuscript since the results were outlined in text.

 5 year mortality 10 year mortality

 NRI (%) 95% CI of NRI p for difference NRI (%) 95% CI of NRI p for difference

age<70 22.3 (10.6, 34.0) 0.047 4.6 (-2.5,11.8) 0.654

age>=70 9.3 ( 4.2, 14.4) 2.9 ( 0.3, 5.4) 

Women 7.4 (1.4, 13.4) 0.049 4.4 ( 1.2 ,7.6) 0.965

Men 16.9 (9.6, 24.3) 4.5 (-0.2, 9.2) 

White 8.8 (0.7, 16.9) 0.977 0.8 (-3.5, 5.2) 0.187

Black 12.1 (4.1, 20.0) 6.5 ( 1.9,11.2) 

Hispanic 14.9 (6.6, 23.2) 4.9 ( 0.2, 9.7) 

We have included the AUC’s in our figures to describe overall model fit in these groups. In conclusion serum cystatin-C based GFR estimated performed better in those under age 70 and in men. Overall however the AUC was low for both cystatin-C and creatinine based GFR estimations emphasizing the need for further better data in diverse populations such as the elderly and women. We have now included this as an additional comment in the results and discussion.

“Interestingly in our cohort the predictive ability of eGFR (regardless of serum measure) appeared higher in the younger participants and men who were most likely to be included in prior cohort that derived GFR estimation formulae. These results highlight the importance of improved accuracy in measurement of GFR in diverse populations will help better understand how CKD is associated with CVD mortality related disparities.”

Reviewer #2:

1. I believe that authors knew serum cystatin C concentration is influenced by many factors, hyper/hypo thyroid, HIV infection, and so on. I could not find in the manuscript about exclusion of these patient from cohort.

Response: 

We thank the reviewer for this comment. We did not collect information on thyroid and HIV status is NOMAS, but these conditions were not exclusion criteria for the Northern Manhattan Study. Participants were excluded from a medical condition perspective only if they already had a stroke.

2. Main finding of this study may be “eGFRcys predicted all-cause mortality better than eGFRcre”. However, this fact is already reported indirectly as the authors cited in Ref 24, 33,34, and Astor BC et al 2011, Peralta CA et al 2011

Response:

We agree with the reviewer, however this topic has not been explored to the same degree in diverse, multi-ethnic, and more predominantly older populations such as the Northern Manhattan Study.

---

## [Decision Letter · Decision Letter 1]

15 Nov 2019

PONE-D-19-20793R1

Creatinine versus Cystatin C for Renal Function-Based Mortality Prediction in an Elderly Cohort: the Northern Manhattan Study

PLOS ONE

Dear Dr Willey,

Thank you for submitting your manuscript to PLOS ONE. After careful consideration, we feel that it has merit but does not fully meet PLOS ONE’s publication criteria as it currently stands. Therefore, we invite you to submit a revised version of the manuscript that addresses the points raised during the review process.

As a reviewer pointed out, cystatin C is affected by multiple conditions. It is a limitation of this study that you can not exclude those cohort with thyroid dysfunction, HIV infection and others.  The authors should describe the limitation on this point.

We would appreciate receiving your revised manuscript by Dec 30 2019 11:59PM. To enhance the reproducibility of your results, we recommend that if applicable you deposit your laboratory protocols in protocols.io, where a protocol can be assigned its own identifier (DOI) such that it can be cited independently in the future. For instructions see: http://journals.plos.org/plosone/s/submission-guidelines#loc-laboratory-protocols

We look forward to receiving your revised manuscript.

Kind regards,

Tatsuo Shimosawa, M.D., Ph.D.

Academic Editor

PLOS ONE

Reviewers' comments:

Reviewer's Responses to Questions

**Comments to the Author**

1. If the authors have adequately addressed your comments raised in a previous round of review and you feel that this manuscript is now acceptable for publication, you may indicate that here to bypass the “Comments to the Author” section, enter your conflict of interest statement in the “Confidential to Editor” section, and submit your "Accept" recommendation.

Reviewer #1: All comments have been addressed

Reviewer #2: (No Response)

2. Is the manuscript technically sound, and do the data support the conclusions?

Reviewer #1: Yes

Reviewer #2: Partly

3. Has the statistical analysis been performed appropriately and rigorously? 

Reviewer #1: Yes

Reviewer #2: Yes

4. Have the authors made all data underlying the findings in their manuscript fully available?

Reviewer #1: Yes

Reviewer #2: Yes

5. Is the manuscript presented in an intelligible fashion and written in standard English?

Reviewer #1: Yes

Reviewer #2: Yes

6. Review Comments to the Author

Reviewer #1: The revised manuscript by Wiiley et al. responded well to the points raised. I have no further critique.

Reviewer #2: At least authors should refer in manuscript about my previous comment 1. Because it must exist and affect on the results.

7. PLOS authors have the option to publish the peer review history of their article (what does this mean?). If published, this will include your full peer review and any attached files.

Reviewer #1: No

Reviewer #2: No

---

## [Author Response · Author response to Decision Letter 1]

26 Nov 2019

We appreciate the reviewer’s comments on our manuscript. We have included the comments by the reviewers below and have modified the manuscript as requested in the appropriate sections and included below are responses to the reviews.

Reviewer: 

1. As a reviewer pointed out, cystatin C is affected by multiple conditions. It is a limitation of this study that you can not exclude those cohort with thyroid dysfunction, HIV infection and others. The authors should describe the limitation on this point. 

and

At least authors should refer in manuscript about my previous comment 1. Because it must exist and affect on the results.

Response:

We agree with the reviewer and editor that the lack of this kind of medical comorbidity information is a limitation of our study and have included the following sentence in the limitation sections:

“Cystatin-C levels can be affected by several medical conditions including thyroid dysfunction44 and human immunodeficiency virus infection45 which unfortunately we did not collect in NOMAS.”

We have also added the very helpful references by the reviewer on other studies that have studied creatinine and cystatin as predictors (Astor BC et al 2011, Peralta CA et al 2011).

---

## [Editor Report · Decision Letter 2]

2 Dec 2019

Creatinine versus Cystatin C for Renal Function-Based Mortality Prediction in an Elderly Cohort: the Northern Manhattan Study

PONE-D-19-20793R2

Dear Dr. Willey,

We are pleased to inform you that your manuscript has been judged scientifically suitable for publication and will be formally accepted for publication once it complies with all outstanding technical requirements.

With kind regards,

Tatsuo Shimosawa, M.D., Ph.D.

Academic Editor

PLOS ONE
---

## [Editor Report · Acceptance letter]

2 Jan 2020

PONE-D-19-20793R2 

Creatinine versus Cystatin C for Renal Function-Based Mortality Prediction in an Elderly Cohort: the Northern Manhattan Study 

Dear Dr. Willey:

I am pleased to inform you that your manuscript has been deemed suitable for publication in PLOS ONE. Congratulations! Your manuscript is now with our production department. 

With kind regards,

on behalf of

Prof. Tatsuo Shimosawa 

Academic Editor

PLOS ONE